# Dual histone methyl reader ZCWPW1 facilitates repair of meiotic double strand breaks in male mice

Mohamed Mahgoub[1†], Jacob Paiano[2,3†], Melania Bruno[1], Wei Wu[2], Sarath Pathuri[4], Xing Zhang[4], Sherry Ralls[1], Xiaodong Cheng[4], André Nussenzweig[2], Todd S Macfarlan[1*]

[1]The Eunice Kennedy Shriver National Institute of Child Health and Human Development, NIH, Bethesda, United States; [2]Laboratory of Genome Integrity, National Cancer Institute, NIH, Bethesda, United States; [3]Immunology Graduate Group, University of Pennsylvania, Philadelphia, United States; [4]Department of Epigenetics and Molecular Carcinogenesis, University of Texas MD Anderson Cancer Center, Houston, United States

**Abstract** Meiotic crossovers result from homology-directed repair of DNA double-strand breaks (DSBs). Unlike yeast and plants, where DSBs are generated near gene promoters, in many vertebrates DSBs are enriched at hotspots determined by the DNA binding activity of the rapidly evolving zinc finger array of PRDM9 (PR domain zinc finger protein 9). PRDM9 subsequently catalyzes tri-methylation of lysine 4 and lysine 36 of Histone H3 in nearby nucleosomes. Here, we identify the dual histone methylation reader ZCWPW1, which is tightly co-expressed during spermatogenesis with *Prdm9*, as an essential meiotic recombination factor required for efficient repair of PRDM9-dependent DSBs and for pairing of homologous chromosomes in male mice. In sum, our results indicate that the evolution of a dual histone methylation writer/reader (PRDM9/ ZCWPW1) system in vertebrates remodeled genetic recombination hotspot selection from an ancestral static pattern near genes towards a flexible pattern controlled by the rapidly evolving DNA binding activity of PRDM9.

*For correspondence:
todd.macfarlan@nih.gov

†These authors contributed equally to this work

Competing interests: The authors declare that no competing interests exist.

## Introduction

Meiotic recombination generates genetic diversity in sexually reproducing organisms and facilitates proper synapsis and segregation of homologous chromosomes in gametes. During meiotic prophase I, recombination is initiated by programmed double strand breaks (DSBs) in DNA at thousands of specific 1–2 kb regions called hotspots (*Kauppi et al., 2004*). DSBs are repaired as either crossovers or non-crossovers. At crossovers, DSBs are repaired using the homologous chromosome as a template, which is critical for homolog pairing, synapsis, and the successful completion of meiosis (*Baudat et al., 2013*; *Zickler and Kleckner, 2015*). In many species, hotspots are distinguished by the presence of active histone marks in chromatin. For example, in yeast, plants, and birds, hotspots are located at regions enriched with histone H3 tri-methylated on lysine 4 (H3K4me3), typically at gene promoters (*Choi et al., 2013*; *Lam and Keeney, 2015*; *Lichten, 2015*; *Singhal et al., 2015*). In contrast, in mammals hotspots are determined by the DNA binding zinc finger array of PRDM9 (PR domain zinc finger protein 9) (*Baudat et al., 2010*; *Myers et al., 2010*; *Parvanov et al., 2010*). PRDM9 catalyzes tri-methylation of lysine 4 and lysine 36 of Histone 3 (H3K4me3 and H3K36me3 respectively) in nearby nucleosomes (*Eram et al., 2014*; *Powers et al., 2016*; *Wu et al., 2013*). This methyltransferase activity is essential for localization of DSBs at PRDM9 binding sites (*Diagouraga et al., 2018*). *Prdm9* loss-of-function in *Mus musculus* leads to sterility (*Hayashi et al.,*

2005), and *Prdm9* heterozygosity in some hybrid mice causes sterility (*Davies et al., 2016*), making *Prdm9* the only known speciation gene so far identified in mammals (*Mihola et al., 2009*). Furthermore, single nucleotide polymorphisms in PRDM9 have been linked to non-obstructive azoospermia in humans (*Irie et al., 2009*; *Miyamoto et al., 2008*).

During meiotic prophase I, chromosomes re-organize as arrays of DNA loops stemming from an axis composed of protein complexes (*Cohen and Holloway, 2014*; *Xu et al., 2019*). Meiotic recombination occurs simultaneously with homologous chromosome pairing and synapsis, and the relationship between these events is complex (*Gray and Cohen, 2016*; *Santos, 1999*; *Zickler and Kleckner, 2015*). Recombination is achieved via homologous repair of programmed DSBs at hotspots. DSBs are initiated by SPO11 protein binding and DNA nicking at recombination hotspot sites (*Bergerat et al., 1997*; *Keeney, 2008*; *Keeney et al., 1997*; *Lange et al., 2016*), followed by DNA end resection. Single strand DNA binding proteins DMC1 and RAD51 are then recruited to facilitate homologous strand invasion, formation of Holliday junctions, and subsequent resolution as either a crossover or non-crossover (*Baudat et al., 2013*). Synapsis is achieved simultaneously with meiotic recombination by connecting the two proteinaceous cores of each homolog axis through a central region containing the synaptic protein SYCP1, which self assembles via its N-terminus, facilitating the closure of the synaptonemal complex (SC) like a zipper (*Fraune et al., 2012*). In mice, a fully assembled SC is required for later steps in recombination and crossovers (*de Vries et al., 2005*; *Hamer et al., 2008*). Thus, the formation of the SC may facilitate meiotic recombination by physically connecting the two chromosomes. Likewise, SC formation and proper synapsis requires meiotic recombination machinery including Dmc1, Rad51, and Mre11 (*Buis et al., 2008*; *Dai et al., 2017*; *Yoshida et al., 1998*).

Despite PRDM9's role in specifying hotspots in mice, DSBs are still produced in *Prdm9* knock-out (KO) mice, but they are re-positioned to the 'default' position at promoters (*Brick et al., 2012*). However, these DSBs are not repaired efficiently, leading to meiotic arrest and partial asynapsis. In F1 hybrid mice, the presence of symmetric PRDM9 binding motifs on both homologs improves recombination rates (*Hinch et al., 2019*; *Li et al., 2019b*), suggesting that PRDM9 itself, and perhaps its downstream factors, play important roles beyond merely initiating DSBs.

*Prdm9* first evolved in jawless vertebrates, and it has been either completely or partially lost in several lineages of vertebrates, including several fish lineages, birds, crocodiles, and canids, indicating it is not absolutely required for meiotic recombination, synapsis, and fertility (*Baker et al., 2017*). Furthermore, *Prdm9* KO mice crossed into other strain backgrounds partially restores fertility in male mice (*Mihola et al., 2019*). Nonetheless, the evolution of PRDM9 in vertebrates replaced a pre-existing hotspot selection system based on the presence of single H3K4me3 marks at promoters with a new selection system based on the presence of dual H3K4me3/H3K36me3 marks at PRDM9 binding sites. In yeast, the histone reader Spp1 links H3K4me3 sites at promoters with the meiotic recombination machinery, promoting DSB formation (*Adam et al., 2018* ). However, in mice, the Spp1 orthologue CXXC1 which also interacts with PRDM9 is not essential for DSB generation or meiotic recombination (*Tian et al., 2018*). It remains unknown whether species with PRDM9 (like mammals) evolved a specialized histone reader (equivalent for Spp1 in yeast) to recognize the dual histone marks catalyzed by PRDM9. Here, we identify Zinc Finger CW-Type and PWWP Domain Containing 1 (ZCWPW1) as a dual histone methylation reader specific for PRDM9 catalyzed histone marks (H3K4me3 and H3K36me3) that facilitates the repair of PRDM9-induced DSBs. Our study reveals a novel histone methylation writer/reader system that controls patterns of meiotic recombination in mammals.

## Results

### ZCWPW1 is a dual histone methylation reader co-expressed with PRDM9 in spermatocytes

To identify PRDM9 co-factors that may play a role in meiotic recombination, we searched for *Prdm9* co-expressed genes in single cells during meiosis from a published dataset (*Chen et al., 2018*). As expected, the top *Prdm9*-correlated genes are known factors in spermatogenesis or DNA metabolism (*Figure 1A*, *Figure 1—source data 1*), however the most correlated gene in our analysis was *Zcwpw1* (rho = 0.519, p-value=2E-6). Similar to *Prdm9*, *Zcwpw1* mRNA is highly expressed

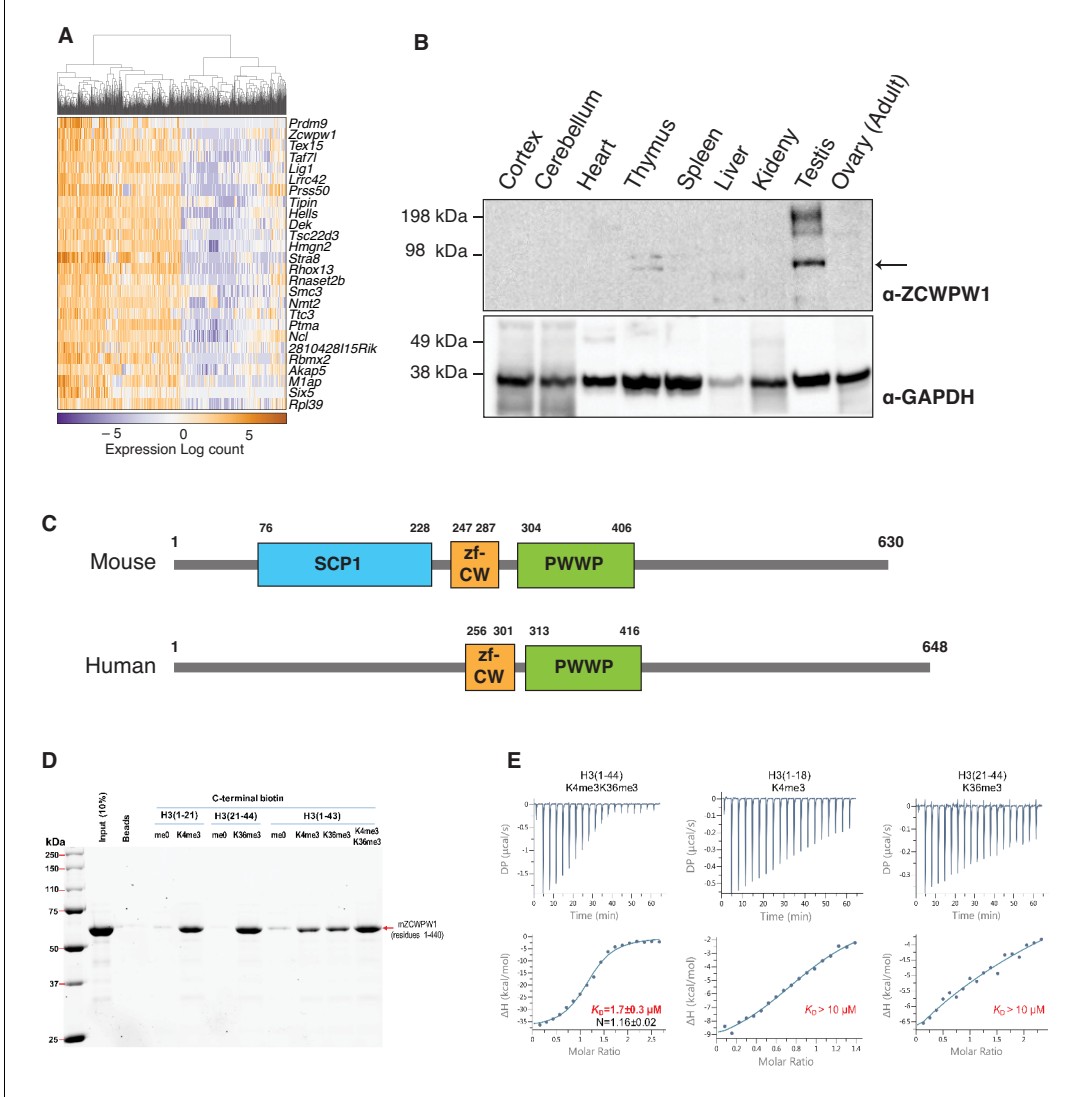

**Figure 1.** ZCWPW1 tissue expression and methyl histone binding activity in vitro. (**A**) Heatmap of the top 25 genes co-expressed with PRDM9 during meiosis in single cells (*Chen et al., 2018*). Rows (genes) are ordered based on PRDM9 correlation coefficient (*rho*), from *Figure 1—source data 1*. Columns (single cells) were clustered by hierarchical clustering using euclidean distance and complete linkage. (**B**) Western blot for endogenous ZCWPW1 protein expression in different tissues from WT B6 mice. The arrow indicates ZCWPW1 band. (**C**) Schematic representation of mouse and human ZCWPW1 proteins with structural domains shown as annotated from NCBI Conserved Domain Database. (**D**) Recombinant ZCWPW1 was mixed with indicated biotinylated histone peptides and bound proteins were identified by Coomassie staining. (**E**) Isothermal titration calorimetry measurements were performed with recombinant ZCWPW1 and indicated histone peptides. The top panel is the raw data showing the heat released and measured by the sensitive calorimeter during gradual titration of the peptide into the sample cell containing ZCWPW1 until the binding reaction has reached equilibrium. The bottom panel shows each peak in the raw data is integrated and plotted versus the molar ratio of peptide to protein. The resulting isotherm can be fitted to a binding model from which the affinity ($K_D$) is derived. The Y-axis measures enthalpy change ($\Delta H$) using the relationship $\Delta G = \Delta H - T \Delta S$ where $\Delta G$ is the Gibbs free energy, $\Delta S$ is the entropy change and T is the absolute temperature. N refers to the reaction stoichiometry.

The online version of this article includes the following source data and figure supplement(s) for figure 1:

**Source data 1.** Correlation of expression between cellular genes and *Prdm9* in single cell RNA-seq analysis.

**Figure supplement 1.** *Zcwpw1* transcript expression in human and mouse tissues.

**Figure supplement 2.** ZCWPW1 antibody specificity.

exclusively in the testis in both mouse and human (*Figure 1—figure supplement 1A–B*). To detect endogenous expression of ZCWPW1 protein, we generated polyclonal rabbit antibodies against full-length ZCWPW1. We confirmed that ZCWPW1 expression is mostly restricted to the testis in mice (*Figure 1B*). We also ruled out the possibility of cross reactivity of our antibody with ZCWPW2, a paralog for ZCWPW1 (*Liu et al., 2016*; *Figure 1—figure supplement 2*). Mouse ZCWPW1 has three recognizable domains from the NCBI conserved domain (CD) database: SCP1, zf-CW and PWWP (*Figure 1C*). The PWWP domain found in multiple proteins binds specifically to histone H3 containing the H3K36me3 mark (*Qin and Min, 2014*), while the zf-CW domain of ZCWPW1 was found to possess H3K4me3-specific binding activity (*He et al., 2010*). The SCP1 domain has homology to the synaptonemal complex protein 1 (SYCP1), the major component of the transverse filaments of the synaptonemal complex (*de Vries et al., 2005*). The N-terminal region of SYCP1 homologous to the SCP1 domain of ZCWPW1 plays a role in SYCP1 dimerization that facilitates synaptonemal complex assembly (*Seo et al., 2016*). It is notable that the NCBI CD database does not recognize the SCP1-like domain in other ZCWPW1 orthologs, including human (*Figure 1C*).

PRDM9 catalyzes the formation of both H3K4me3 (*Hayashi et al., 2005*; *Wu et al., 2013*) and H3K36me3 (*Eram et al., 2014*; *Powers et al., 2016*) on the same histone molecules in vitro and the same nucleosomes in vivo (*Powers et al., 2016*). Therefore, we reasoned that ZCWPW1 may link the PRDM9-induced histone marks to the meiotic recombination machinery by binding to the dually marked histone tails. Using in vitro biotin-streptavidin pulldown assays, we determined that recombinant ZCWPW1 (residues 1–440) binds with higher affinity to H3K4me3 and H3K36me3 than non-methylated biotinylated H3 peptides (*Figure 1D*). Importantly, dual-modified H3K4me3/K36me3 peptides had the highest binding affinity for ZCWPW1 compared to peptides with either single modification (*Figure 1D*). We further quantified the binding of ZCWPW1 to histone peptides by Isothermal Titration Calorimetry, which demonstrated that ZCWPW1 binds with the highest affinity to H3K4me3/K36me3 peptides ($K_D$ = 1.7 ± 0.3 μM) compared to H3K4me3 or H3K36me3 alone ($K_D$ >10 μM each), with a 1:1 stoichiometry (*Figure 1E*). In sum these results indicate that ZCWPW1 is a meiosis specific histone methylation reader for the unique dual methyl marks catalyzed by PRDM9, suggesting that ZCWPW1 and PRDM9 have complementary roles in meiosis.

## Co-evolution of *Zcwpw1* and *Prdm9* in vertebrates

PRDM9 has an extraordinary evolutionary pattern as it possesses a rapidly evolving zinc finger array leading to distinct hotspots even among species belonging to the same genus (*Baker et al., 2017*; *Oliver et al., 2009*; *Thomas et al., 2009*). Furthermore, while it emerged in jawless fishes, it has been repeatedly lost in several species across different clades including in birds, crocodiles, and amphibians, as well as in several lineages of fish. To determine the evolutionary pattern of *Zcwpw1*, we retrieved *Zcwpw1* orthologs from three databases: (1) NCBI HomoloGene, (2) Ensembl and (3) OrthoDB (*Kriventseva et al., 2019*). We then performed a motif analysis in these orthologs using the ScanProsite tool (*de Castro et al., 2006*) and NCBI CD search, and excluded orthologs lacking a zf-CW or PWWP domain. Using this strategy, we identified 174 vertebrate species with ZCWPW1 (*Figure 2—source data 1*). To find additional *Zcwpw1* orthologs, we extended our search by performing BLAST (blastp and tblastn) searches using the NCBI nr and nt databases respectively against all vertebrate genomes. We used sequences of zf-CW and PWWP domains from mouse ZCWPW1 as a query for our search. BLAST hits were then scanned for presence of both zf-CW and PWWP domains, and only hits with the two domains present were considered for further analysis. In addition to ZCWPW1, our analysis uncovered orthologs for ZCWPW2, a ZCWPW1 paralog which also contains zf-CW and PWWP domains and diverged from ZCWPW1 in the last common ancestor of vertebrates. To differentiate between ZCWPW1 and ZCWPW2 orthologs in our BLAST results, we added known orthologs for both proteins (ZCWPW1 and ZCWPW2) to our candidate sequences from BLAST, and performed a multiple sequence alignment and built a maximum likelihood phylogenetic tree. ZCWPW1 and ZCWPW2 orthologs clustered into two distinct branches, and their branching node was supported by a bootstrap value of 99% (*Figure 2—figure supplement 1*). BLAST hits which clustered with ZCWPW1 are considered as novel orthologs in our analysis. Using this approach, we identified an additional 19 species with ZCWPW1 orthologs (*Figure 2—source data 1*). Next, we looked for sequence conservation of ZCWPW1 by performing multiple sequence alignment of all identified orthologs. Both the zf-CW and PWWP domains are highly conserved (*Figure 2A*, *Figure 2B*, *Figure 2—figure supplement 2*). Furthermore, the N-terminal region

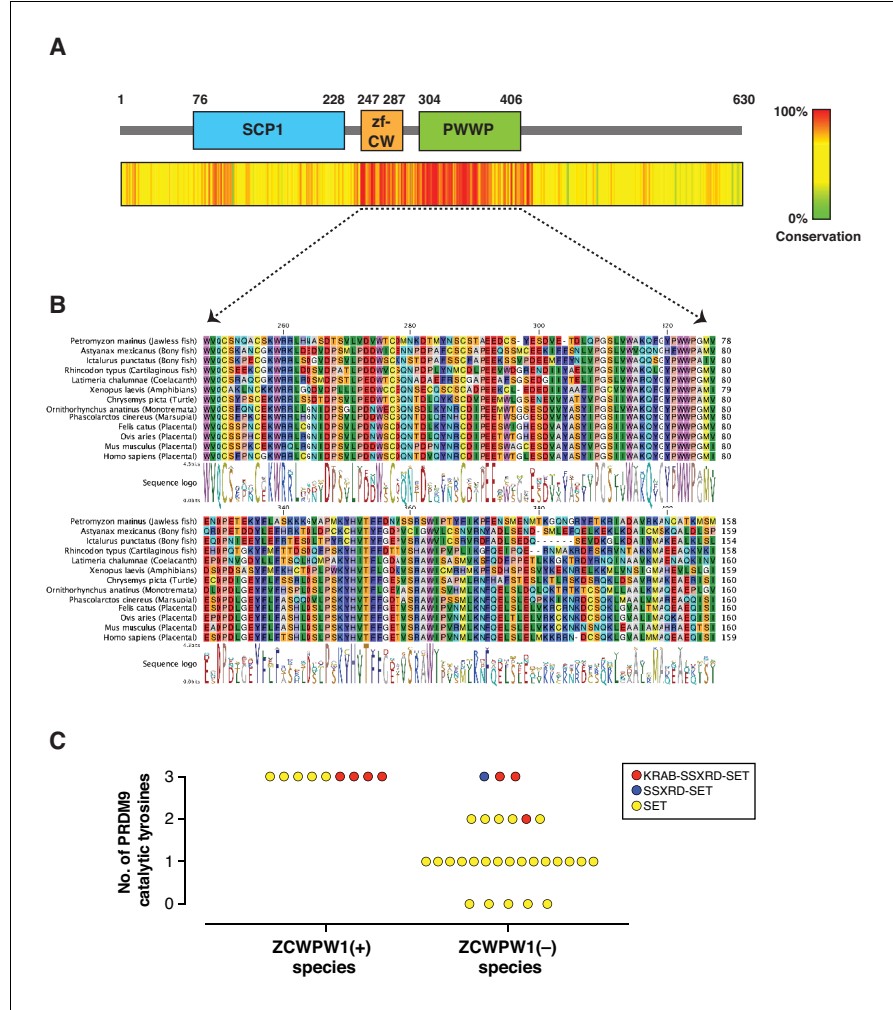

**Figure 2.** Evolution of *Zcwpw1* in vertebrates. (**A**) ZCWPW1 protein sequences available from public databases (174 species) were aligned by CLC Genomics Workbench using MUSCLE algorithm (*Edgar, 2004*). Alignment conservation score is plotted as heat map across different regions of mouse ZCWPW1 shown in the schematic figure above the heatmap. Alignment segments corresponding to gaps in the reference sequence (mouse ZCWPW1) were removed. (**B**) Multiple sequence alignment from (**A**) showing regions corresponding to zf-CW and PWWP domains in 13 species representing different vertebrate clades. Consensus sequence logo is shown below the alignment. (**C**) Plot showing number of conserved catalytic tyrosines (n = 3) in the SET domain of PRDM9 in bony fish species that have a PRDM9 ortholog. PRDM9 orthologs are divided into two groups depending on whether the corresponding species of that ortholog has ZCWPW1 ortholog (with ZCWPW1 group) or lacks ZCWPW1 ortholog (without ZCWPW1 group). Different colors represent different domain architecture of PRDM9 ortholog in the particular species. PRDM9 information reproduced from *Baker et al., 2017*.

The online version of this article includes the following source data and figure supplement(s) for figure 2:

**Source data 1.** ZCWPW1 orthologs used in evolution analysis.
**Figure supplement 1.** Phylogenetic analysis of potential ZCWPW1 orthologs discovered by BLAST.
**Figure supplement 2.** Amino acid variability in ZCWPW1 orthologs.
**Figure supplement 3.** Alignment of SCP1 domains from mouse ZCWPW1 and SYCP1.
**Figure supplement 4.** PRDM9 domain structure in bony fishes.

corresponding to the SCP1 domain in mouse ZCWPW1 showed weaker but detectable conservation. When we aligned the SCP1 domain of mouse ZCWPW1 with SCP1 domain of mouse SYCP1 (*Figure 2—figure supplement 3A*), we observed 30% identity. Furthermore, the amino acids that match between ZCWPW1 and SYCP1 have detectable conservation among ZCWPW1 orthologs (*Figure 2—*

*figure supplement 3B*), albeit less that what is seen in zf-CW and PWWP domains (*Figure 2—figure supplement 2*).

Like *Prdm9*, *Zcwpw1* emerged in jawless fishes and is present in some bony and in cartilaginous fishes. Interestingly, we could not identify any birds or crocodile species with *Zcwpw1* orthologs, similar to *Prdm9* (*Baker et al., 2017*). However, unlike *Prdm9,* some amphibians and dogs have *Zcwpw1* orthologs (*Figure 2—source data 1*). Bony fishes represent a heterogenous group in regard to *Prdm9* evolution, as there have been multiple events of *Prdm9* loss and they have variable PRDM9 domain structures (*Baker et al., 2017*; *Figure 2—source data 1*). According to our analysis, several bony fish species with *Prdm9* contain *Zcwpw1* orthologs (n = 9), while the majority lack *Zcwpw1* orthologs (n = 29) (*Figure 2—figure supplement 4*). Also, we did not detect any association between PRDM9 domain structure and the presence of a *Zcwpw1* ortholog. Next, we looked for the conservation of the catalytic tyrosine residues of PRDM9 SET domain (Y276, Y341 and Y357), which are critical for PRDM9 methyltransferase activity (*Wu et al., 2013*). Strikingly, we found that all species with both *Prdm9* and *Zcwpw1* orthologs maintained the three catalytic tyrosine residues of PRDM9, whereas the three catalytic tyrosines were present in only 3 out of 29 species lacking *Zcwpw1* orthologs (*Figure 2C*). These findings are similar to the findings of a recent report (*Wells et al., 2019*). This co-evolutionary pattern of *Prdm9* catalytic tyrosines and *Zcwpw1* in bony fishes, as well as the concomitant loss of both *Prdm9* and *Zcwpw1* in birds and crocodiles might imply that, at least in some species, the two genes are essential in a cooperative manner.

## ZCWPW1 binds specifically to dual methylated nucleosomes at meiotic hotspots

Meiotic recombination DSB hotspots in mice are primarily determined by PRDM9 binding (*Baudat et al., 2010*; *Davies et al., 2016*; *Grey et al., 2017*; *Myers et al., 2010*; *Parvanov et al., 2010*). In addition to the C2H2 zinc finger domain, PRDM9 contains a KRAB, SSXRD, and PR-SET domains which are critical for its function (*Diagouraga et al., 2018*; *Imai et al., 2017*; *Thibault-Sennett et al., 2018*). DSB hotspots are characterized by PRDM9-dependent histone H3 methylation (H3K4me3 and H3K36me3) (*Baker et al., 2014*; *Grey et al., 2018*; *Grey et al., 2011*; *Hayashi et al., 2005*; *Smagulova et al., 2011*). To determine whether ZCWPW1 binds to these chromatin sites in vivo, we performed Cleavage Under Targets and Release Using Nuclease (CUT&RUN) (*Christoph and Siegenthaler, 2016*) using custom polyclonal ZCWPW1 antibodies on mouse spermatocytes. We identified 4,487 ZCWPW1 peaks (p<0.001). Most ZCWPW1 peaks overlap with previously identified hotspots defined by the presence of the single-strand (ss) DNA binding protein DMC1 (*Brick et al., 2018*; *Figure 3A*) or released SPO11-oligos (*Lange et al., 2016*; *Figure 3—figure supplement 1A*) (89% and 58% respectively). Additionally, ZCWPW1 signal exhibits positive correlation with SPO11-oligo density, DMC1-SSDS and PRDM9 binding signals (*Figure 3—figure supplement 2*). PRDM9 binding generates a nucleosome-depleted region with nucleosomes organized symmetrically around PRDM9 binding sites (*Baker et al., 2014*). When centering our CUT&RUN signals around PRDM9 motifs, we also found a central region which lacks both ZCWPW1 and H3K4me3 signals. This H3K4me3/ZCWPW1 depleted region is surrounded by symmetrically distributed peaks of H3K4me3 and ZCWPW1 indicative of nucleosome positioning around the hotspots' central nucleosome depleted region (*Figure 3A*, *Figure 3—figure supplement 1B*). It is notable that DSB hotspots that fail to overlap with ZCWPW1 peaks still show ZCWPW1 signal that is below the peak detection threshold (*Figure 3—figure supplement 3A*). Next, we looked for ZCWPW1 overlap with functional regions in the genome: Transcription Start Sites (TSS), Transcription End Sites (TES) and CpG islands (CGI), and we found only 8%, 9% and 5% of our ZCWPW1 called peaks overlap with these sites respectively (*Figure 3—figure supplement 3B*). Despite ZCWPW1 peaks overlapping with these sites, there was no significant ZCWPW1 enrichment genome-wide in these functional sites compared to IgG control (*Figure 3—figure supplement 3A*, *Figure 3—figure supplement 3C*). The weak ZCWPW1 recruitment around transcriptionally active regions (rich in both H3K4me3 and H3K36me3 marks which rarely overlap) suggests that the mere presence of either of these modifications alone is not associated with recruitment of ZCWPW1 in spermatocytes. To investigate whether the presence of both H3K4me3 and H3K36me3 marks is associated with higher ZCWPW1 recruitment to chromatin in vivo, we used a published ChIP-seq dataset of H3K4me3 and H3K36me3 binding in prophase I spermatocytes (*Lam et al., 2019*). While ZCWPW1 binding strength showed positive correlation with H3K4me3 signals, the correlation with

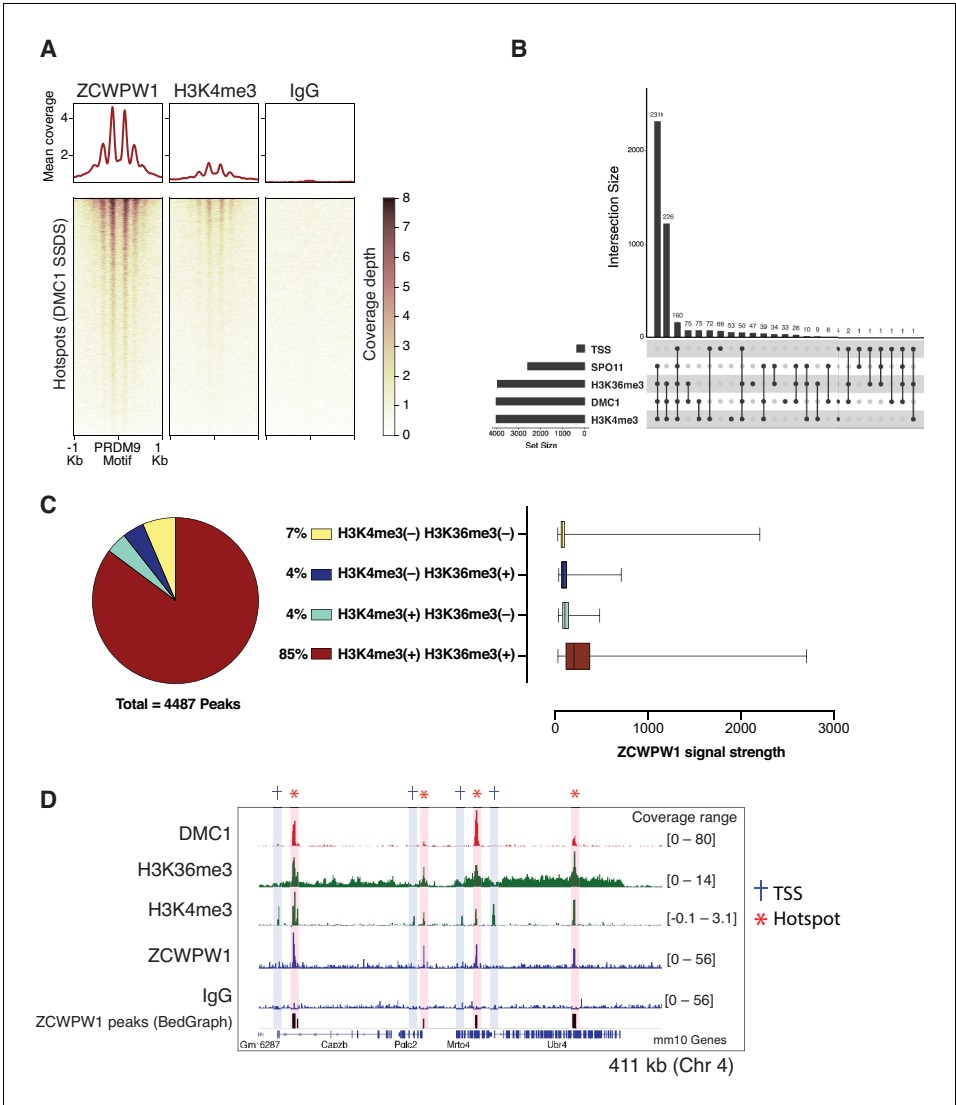

**Figure 3.** Mapping of ZCWPW1 chromatin biding in vivo using CUT&RUN in spermatocytes from B6/B6 mice. (**A**) Heatmaps representing ZCWPW1, H3K4me3 and IgG (anti-GFP) CUT&RUN read coverage in B6/B6 mouse spermatocytes at hotspots determined by DMC1 SSDS (GSE99921). Signals are centered around PRDM9 motifs. Hotspots with multiple motifs were excluded from plotting (11,350 DSB hotspots were used in total). Y-axis of line plots in the top represents mean coverage (trimmed fragments/50 bp) (**B**) Upset plot showing intersections of ZCWPW1 peaks (n = 4,487) with SPO11 oligos (GSE84689), DMC1 SDDS (GSE99921), prophase H3K4me3 and H3K36me3 peaks (GSE121760) and transcription starting sites (TSS). Y-axis represents number of ZCWPW1 peaks intersecting with the specified regions. (**C**) Left: pie chart showing tri-methylation states of lysine 4 and 36 of histone H3 at ZCWPW1 peaks in B6/B6 spermatocytes. Right: ZCWPW1 signal strength at different ZCWPW1 peak subsets shown in the pie chart on the left. X-axis represents the strength of ZCWPW1 signal (read coverage) per peak. (**D**) Read coverage plots for DMC1 SSDS hotspots (GSE35498/red), H3K4me3/H3K36me3 (GSE121760/Green) and ZCWPW1/IgG (CUT&RUN/blue) across a region on chromosome 4. Hotspots and TSSs are highlighted. Y-axis represents coverage (fragments/bin). The track in the bottom represents the bedGraph of ZCWPW1 called peaks using MACS2.

The online version of this article includes the following figure supplement(s) for figure 3:

**Figure supplement 1.** Mapping of ZCWPW1 chromatin biding in vivo using CUT&RUN in spermatocytes from B6/B6 mice.

**Figure supplement 2.** ZCWPW1 correlation with other metrics of meiotic recombination at DSB hotspots.

**Figure supplement 3.** ZCWPW1 binding at functional genomic sites.

**Figure supplement 4.** ZCWPW1 binding at chromosome X pseudoautosomal region.

H3K36me3 was weaker, probably because the H3K36me3 signal at hotspots is notably weaker than that of the other commonly used DSB metrics (*Figure 3—figure supplement 2*). The vast majority of ZCWPW1 peaks overlapped with regions containing both H3K4me3 and H3K36me3 marks (85%), with significantly less peaks overlapping with regions marked by either mark alone (*Figure 3C*, *Figure 3D*). In agreement with in vitro binding assays (*Figure 1D*, *Figure 1E*), we found that ZCWPW1 CUT&RUN signals are significantly stronger in peaks with H3K4me3/H3K36me3 dual marks compared to regions with a single mark (*Figure 3C*). Finally, we did not observe any enrichment of ZCWPW1 relative to control in the pseudo-autosomal region (PAR) of sex chromosomes, which undergoes PRDM9-independent recombination in mice (*Brick et al., 2012*; *Figure 3—figure supplement 4*). Collectively, genome-wide mapping of ZCWPW1 demonstrates its selective binding at PRDM9 determined hotspots at regions containing dual trimethylation of histone H3 lysine 4 and lysine 36.

## ZCWPW1 chromatin occupancy is determined by PRDM9 in allele-specific manner

To determine factors responsible for ZCWPW1 recruitment to chromatin in vivo in an unbiased manner, we searched for enriched motifs within ZCWPW1 peak regions. This analysis identified exclusively the PRDM9 binding motif of the Dom2 allele (PRDM9$^{Dom2}$), found within the C57Bl/6J strain (p=1e-261) (*Figure 4—source data 1*), suggesting that ZCWPW1 occupancy is solely PRDM9-dependent. To test this hypothesis, we mapped ZCWPW1 binding in F1-hybrid mouse spermatocytes from a C57Bl/6J (B6) and CAST/EiJ (CAST) cross, since CAST mice encode a unique PRDM9 allele (PRDM9$^{Cast}$) with distinct DNA binding properties. In these mice, we identified 8,976 peaks, 60% of which overlap with previously described CAST/CAST hotspots (*Smagulova et al., 2016*), while only 18% overlap with B6/B6 hotspots (*Figure 4—figure supplement 1A*), a finding consistent with reported PRDM9$^{Cast}$ dominance in hybrid mice (*Baker et al., 2015*). Furthermore, ZCWPW1 also binds to 1,383 novel hotspots that do not exist in either B6/B6 or CAST/CAST mice, but only appear in F1-hybrids, where PRDM9$^{Cast}$ binds to the B6 genome and PRDM9$^{Dom2}$ binds to the CAST genome (*Figure 4—figure supplement 1A*). These sites have been shown to be due to PRDM9-binding-site erosion that occurs as a result of accumulated gene conversion events within each subspecies (*Baker et al., 2015*). De novo motif discovery in the hybrids revealed two motifs identical to PRDM9$^{Cast}$ and PRDM9$^{Dom2}$ (p=1e-768 and 1e-60 respectively) (*Figure 4A*; *Figure 4—source data 2*). This resemblance between ZCWPW1 and PRDM9 in their chromatin binding motifs is reflected in the pattern of chromatin occupancy of ZCWPW1 genome-wide in a PRDM9-allele specific manner in both B6/B6 and B6/CAST mice (*Figure 4B*, *Figure 4—figure supplement 1B*). Overall, these results indicate that ZCWPW1 occupancy is a novel marker for meiotic hotspots that reflects PRDM9 allele specificity, and ZCWPW1 CUT&RUN is a useful method to infer PRDM9 binding sites.

## *Zcwpw*1 knockout mice are azoospermic and display asynapsis and repair defects

PRDM9-induced DSBs are critical for successful chromosome synapsis during meiosis, and *Prdm9*-null male mice (*M. m. domesticus)* are azoospermic as a result of meiotic arrest due to compromised DSB repair and chromosomal asynapsis (*Hayashi et al., 2005*). As ZCWPW1 binding overlapped PRDM9 binding sites, we hypothesized it also plays important role in homolog synapsis during meiosis. To test that hypothesis, we generated mice with homozygous deletion of *Zcwpw1* (*Zcwpw1* KO), which were born healthy and at mendelian ratios. Western blotting confirmed the absence of ZCWPW1 in testis from *Zcwpw1* KO mice (*Figure 5—figure supplement 1A*), which are smaller and have reduced mass (*Figure 5—figure supplement 1B*) compared to wild type (*Zcwpw1* WT) (80 mg vs.190 mg). Hematoxylin and Eosin (H&E) staining of tissue sections demonstrated the total absence of spermatids in the seminiferous tubules of *Zcwpw1* KO testes (*Figure 5A* and *Figure 5—figure supplement 1C*). To look for possible defects in homolog pairing and synapsis, we performed double immunofluorescence staining of spermatocyte spreads with the SC axial element marker SYCP3 (which labels the axis of sister chromatids) and the SC central element marker SYCP1 (which stains the synaptonemal complex formed between paired homologs). SYCP3/SYCP1 staining revealed that *Zcwpw1* KO mice arrest at a pachytene-like stage of prophase I (*Figure 5B*). In *Zcwpw1* KO mice, almost all spermatocytes displayed partial chromosomal asynapsis (*Figure 5C*), with an average of

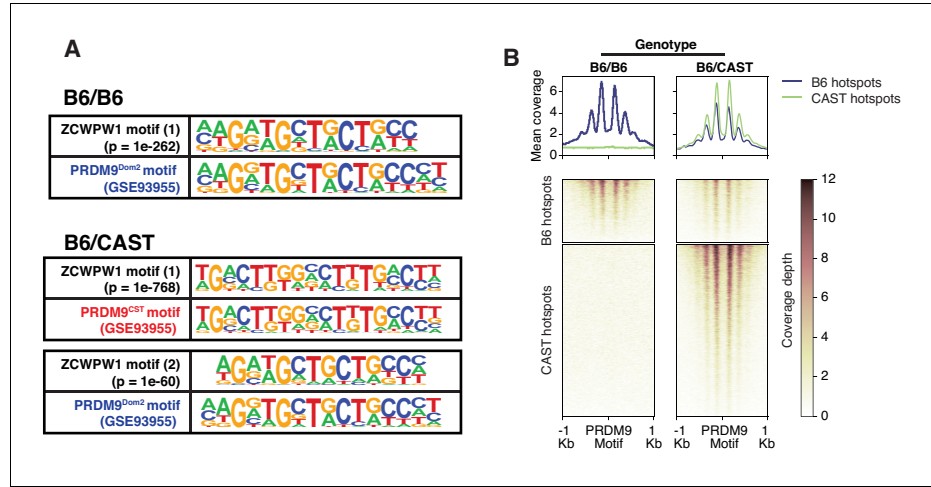

**Figure 4.** Mapping of ZCWPW1 chromatin biding in vivo using CUT&RUN in spermatocytes from F1 B6/CAST hybrid mice. (**A**) Comparisons of de novo discovered motifs for PRDM9$^{Dom2}$, PRDM9$^{Cast}$ and ZCWPW1. For PRDM9$^{Dom2}$ and PRDM9$^{Cast}$, PRDM9 ChIP-seq peaks (GSE93955) for either allele were queried by HOMER to identify the consensus DNA motif for their binding sites. For ZCWPW1, CUT&RUN peaks from either B6/B6 or B6/CAST F1 hybrid were used to identify ZCWPW1 binding motifs in pure and mixed genomic backgrounds, respectively. The first motif hit is shown for binding in B6/B6 and the first and second hits are shown for B6/CAST. (**B**) Heatmaps comparing ZCWPW1 read coverage for CUT&RUN performed in spermatocytes from either B6/B6 or B6/CAST F1 hybrid. ZCWPW1 signal is plotted around F1 hybrid hotspots defined by DMC1 SSDS (GSE73833) and regions categorized based on PRDM9 allele motif in the center of the hotspot (*Prdm9$^{Dom2}$* and *Prdm9$^{Cast}$* for B6 and CAST strains respectively). Signals are centered around PRDM9 motifs, and hotspots with multiple motifs were excluded from plotting (10,466 DSB hotspots from B6xCAST F1 were used in total). Y-axis of line plots in the top represents mean coverage (trimmed fragments/50 bp).

The online version of this article includes the following source data and figure supplement(s) for figure 4:

**Source data 1.** Enrichment of known motifs at ZCWPW1 binding sites in B6/B6 mice.
**Source data 2.** De novo motifs discovery at ZCWPW1 binding sites in B6/B6 and F1 B6/CAST hybrid mice.
**Figure supplement 1.** Mapping of ZCWPW1 chromatin biding in vivo using CUT&RUN in spermatocytes from F1 B6/CAST hybrid mice.

12 autosomal chromosomes synapsed per spermatocyte (out of 19), in contrast to *Zcwpw1* WT spermatocytes which -with rare exceptions- have all homologous autosomal chromosomes fully synapsed in spermatocytes at the pachytene stage (*Figure 5D*).

As synapsis is initiated by PRDM9-induced DSBs, we tested whether asynapsis in spermatocytes from *Zcwpw1* KO mice were due to failure in DSB formation. We stained spermatocytes with antibodies recognizing the DSB marker γ-H2AX and the repair factor DMC1, which is recruited to single stranded DNA resulting from DSBs. Spermatocytes from *Zcwpw1* KO mice have widespread focal accumulation of DMC1 in the cells arrested at pachytene-like stage (*Figure 5E*), with significantly increased numbers of DMC1 foci per spermatocyte in *Zcwpw1* KO (*Figure 5F*). Furthermore, γ-H2AX staining showed significant accumulation on autosomes and the absence of the XY sex body, indicating prophase blockage before sex body formation (*Figure 5—figure supplement 1D*). Our observations from immunofluorescence staining are in agreement with what has been recently reported by two different groups who found that *Zcwpw1* KO *spermatocytes* have pachytene-like meiotic arrest with partial asynapsis and increased DMC1 foci compared to *Zcwpw1* WT *spermatocytes* (*Li et al., 2019a*; *Wells et al., 2019*). These results suggest intact DSB formation by SPO11 but defective repair and synapsis. Collectively, *Zcwpw1* KO mice phenocopy *Prdm9* null male mice in which sperm production is compromised by partially defective DSB repair and synapsis with meiotic arrest at a pachytene-like stage (*Brick et al., 2012*; *Hayashi et al., 2005*).

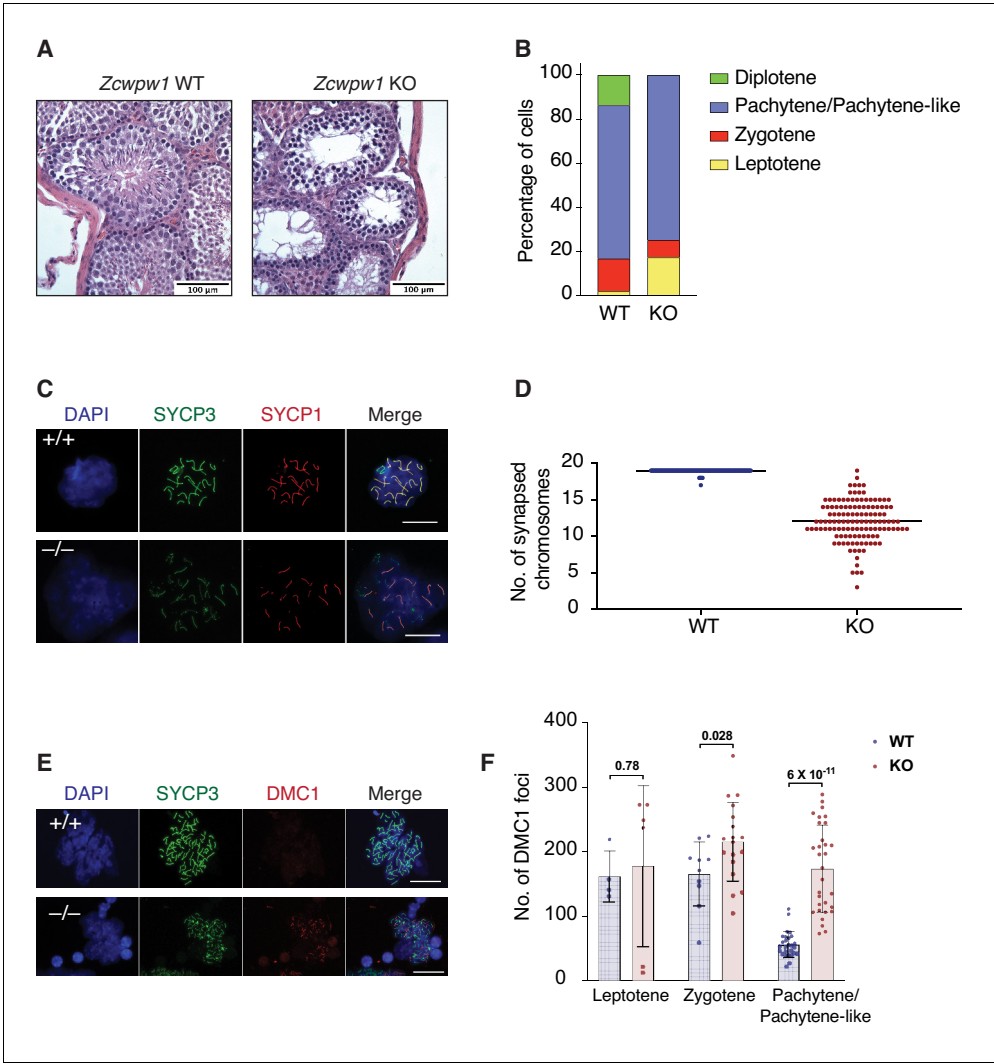

**Figure 5.** *Zcwpw1* KO mice are azoospermic and display hallmarks of defective synapsis and DSB repair. (**A**) Hematoxylin and Eosin staining of paraffin embedded tissue sections from testes of *Zcwpw1* WT or *Zcwpw1* KO. Scale bar is 100 μm. (**B**) Percentage distribution for meiotic prophase I stages in spermatocyte spreads from *Zcwpw1* WT (n = 183) or *Zcwpw1* KO (n = 194). Staging is based on double immunofluorescence staining of SYCP3 and SYCP1. (**C**) Indirect immunofluorescence staining with SYCP3 and SYCP1 in spermatocyte spreads from *Zcwpw1* WT or *Zcwpw1* KO. Scale bar is 10 μm (**D**) Counts for number of fully synapsed homologous chromosomes (autosomes) per spermatocyte. Counting was performed on spermatocyte spreads with double staining of SYCP3 and SYCP1. Only spermatocytes in pachytene/pachytene-like stage are considered for counting in *Zcwpw1* WT or *Zcwpw1* KO (n = 127 for each genotype). (**E**) Indirect immunofluorescence staining with SYCP3 and DMC1 in spermatocyte spreads from *Zcwpw1* WT or *Zcwpw1* KO. Scale bar is 10 μm (**F**) DMC1 foci count per cell in spermatocyte spreads double stained with SYCP3 and DMC1 in *Zcwpw1* WT (Leptotene = 4, Zygotene = 10, Pachytene = 27) or *Zcwpw1* KO (Leptotene = 6, Zygotene = 18, Pachytene-like = 31) spermatocytes. p-value shown is calculated with Welch's t-test.

The online version of this article includes the following figure supplement(s) for figure 5:

**Figure supplement 1.** Phenotyping of *Zcwpw1* KO spermatocytes.

## DSBs are generated at PRDM9-dependent hotspots but are not fully repaired in *Zcwpw1* knockout mice

In *Prdm9*-null male mice, meiotic DSBs reposition away from PRDM9 bound motifs to promoters and at other sites of PRDM9-independent H3K4me3 (*Brick et al., 2012*). As *Zcwpw1* KO mice phenocopy *Prdm9* null mice, we predicted that DSBs in *Zcwpw1* KO mice would be similarly repositioned

to promoters. To map hotspot locations, we utilized END-seq, a quantitative method that sequences DSBs at nucleotide resolution (*Paiano et al., 2020*). We performed END-seq using bulk testicular cells from *Zcwpw1* KO and *Zcwpw1* WT adult mice. Peak calling on END-seq reads identified 2,098 and 3,147 DSB sites in spermatocytes from *Zcwpw1* WT and *Zcwpw1* KO mice respectively. Surprisingly, unlike the case in *Prdm9* KO, DSBs in *Zcwpw1* KO mice nearly completely overlapped with DSBs in *Zcwpw1* WT, and with previously identified hotpots in B6 mice using DMC1 SSDS (*Figure 6A*, *Figure 6B*, *Figure 6C*). Importantly, unlike in *Prdm9* KO mice, DSBs were not repositioned towards promoters (*Figure 6B*).

After normalizing the END-seq signal using spike-in controls (see methods), we observe a distinct END-seq signal profile across all hotspots, with an increased flanking peak signal with relatively unchanged central peak signal (*Figure 6D*). This central peak region largely reflects a recombination intermediate with covalently bound SPO11 resulting from single strand invasion into the unbroken homologous chromosome, whereas the flanking peak regions reflect the extent of end-resection (*Paiano et al., 2020*). This interpretation of central signal is supported by the loss of the central peak at non-PAR region of the X chromosome in *Zcwpw1* WT testes (whose repair does not involve homolog engagement) (*Figure 6D*). The minimal change in the central signal in comparison to end-resection signal might suggest that, in *Zcwpw1* KO spermatocytes, DSB repair is defective downstream of homolog invasion and formation of recombination intermediates. However, given that asynapsis is only partial in *Zcwpw1* KO, and the END-seq signal is representative of both a heterogenous cell population and many hotspots, we can't exclude a potential role of ZCWPW1 in facilitating homolog invasion at some hotspots. In sum these data indicate that unlike PRDM9, ZCWPW1 is not required for the positioning of DSBs, rather it is required exclusively for efficient DSB repair and homologous recombination.

## Discussion

In this study, we used a computational approach to identify PRDM9 co-expressed genes and identified a novel factor, ZCWPW1, that is critical for the repair of PRDM9-induced DSBs during meiosis. Like PRDM9, ZCWPW1 is expressed at very low/undetectable levels in most tissues but becomes expressed at high levels prior to and during meiotic prophase I. *Zcwpw1* is necessary for fertility in male mice, as *Zcwpw1* KO mice are azoospermic, similar to *Prdm9* KO. In contrast, *Zcwpw1* KO females are initially fertile, but suffer from ovarian insufficiency as they age, likely due to delayed meiosis in fetal ovaries (*Li et al., 2019a*). This is also in contrast to *Prdm9* KO females, which are completely infertile (*Hayashi et al., 2005*). These data highlight the distinct checkpoint sensitivities in males and females for meiotic progression defects. Furthermore, the distinct phenotypes of *Prdm9* and *Zcwpw1* loss-of-function in females suggest that PRDM9 has ZCWPW1 independent function in females that will require additional exploration.

PRDM9 is a unique SET domain containing histone methyltransferase in several respects. First, it contains its own specific DNA binding domain, a rapidly evolving C2H2 zinc finger array that binds to target sequences with high specificity. Second, it contains a dual specificity histone methyltransferase activity for histone H3 K4 and histone H3 K36 (*Eram et al., 2014*; *Powers et al., 2016*; *Wu et al., 2013*). Likewise, ZCWPW1 is a unique protein that possesses dual histone methylation reader domains; a zf-CW domain that has previously been shown to bind to the H3K4me3 mark (*He et al., 2010*), and a PWWP domain, which has been shown on multiple proteins including DNMT3a and DNMT3b to bind to the H3K36me3 mark (*Rondelet et al., 2016*). ZCWPW1 can bind to histone H3 peptides with double H3K4me3 and H3K36me3 marks in vitro with high affinity at a 1:1 ratio. Furthermore, PRDM9 predominantly methylates H3K4me3 and H3K36me3 on the +1, +2,–1 and –2 nucleosomes flanking its binding sites, which provides a short platform for ZCWPW1 interaction with chromatin at sites in vivo. Thus, we prefer a model in which ZCWPW1 interacts with both marks on the same H3 peptide, but it cannot be ruled out that ZCWPW1 binds to the two marks on separate H3 molecules on the same or adjacent nucleosomes. Resolving these possibilities will require additional biochemical studies.

The mapping of ZCWPW1 binding to chromatin genome-wide by CUT&RUN demonstrated that the majority of binding sites overlap with meiotic DSB hotspots. Most of these sites are characterized by the concurrent presence of PRDM9-dependent tri-methylation marks: H3K4me3 and H3K36me3. This implies that ZCWPW1 plays a central role as PRDM9 co-factor. Nevertheless, we

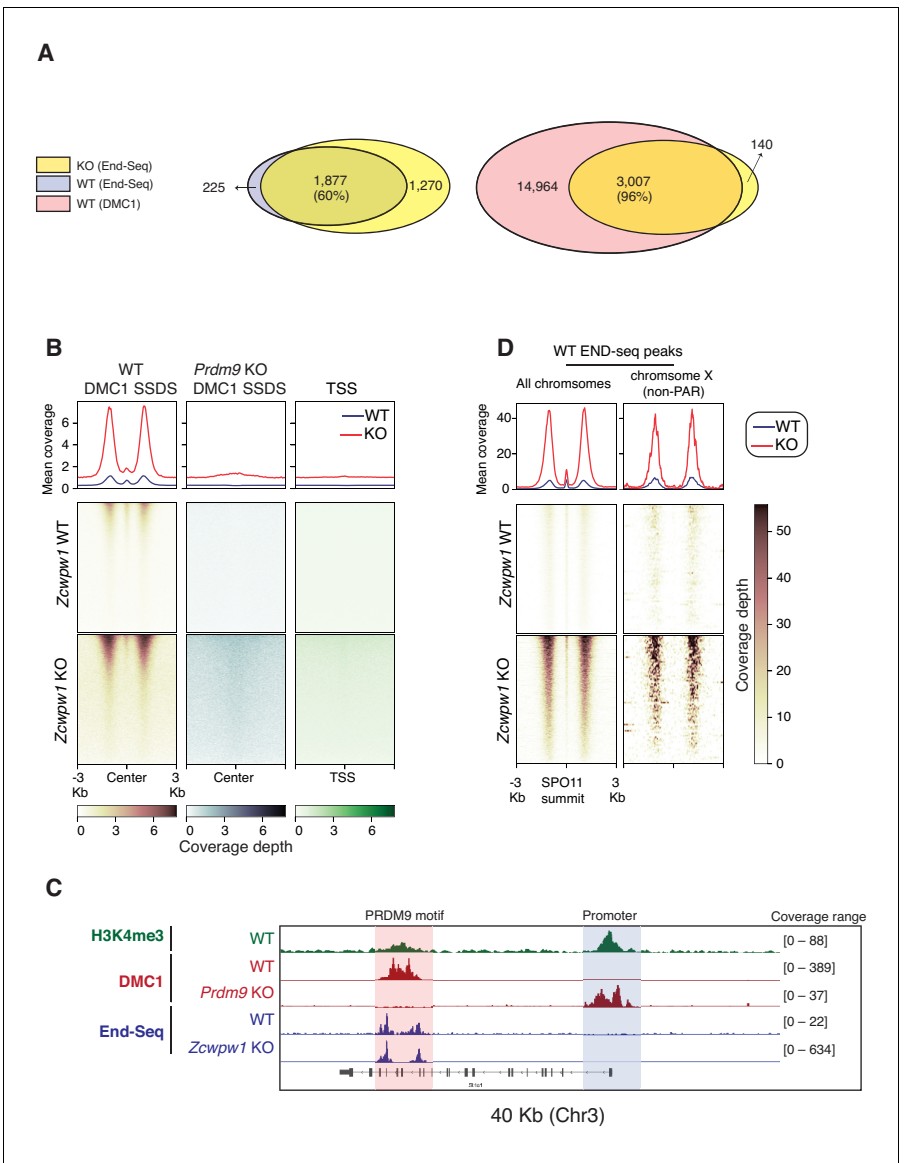

**Figure 6.** Double Strand Break mapping by END-seq in spermatocytes from adult *Zcwpw1* WT and *Zcwpw1* KO mice. (**A**) Venn diagram showing END-seq peak overlap between *Zcwpw1* KO and *Zcwpw1* WT (left) or between *Zcwpw1* KO END-seq peaks and DMC1 SSDS hotspots (GSE35498) (right). (**B**) END-seq read coverage heatmaps comparing *Zcwpw1* WT and *Zcwpw1* KO DSBs at different indicated regions: DMC1 SSDS hotspots (WT and PRDM9 KO/GSE35498) and transcription starting sites (TSS). All signals are normalized with spike-in controls. Y-axis of line plots in the top represents spike normalized Reads Per Million Reads (RPM) (**C**) Read coverage plots for H3K4me3 (CUT&RUN/green), DMC1 SSDS hotspots from WT and PRDM9 KO (GSE35498/red) and END-seq (blue). (**D**) Read coverage heatmaps for END-seq reads from *Zcwpw1* WT and *Zcwpw1* KO spermatocytes. Signal is plotted around WT END-seq called peaks (total peaks on the left, and non-PAR of chromosome X peaks on the right). Signals are centered at the overlapping SPO11 oligo summits (GSE84689). Signal is normalized with spike-in controls. Y-axis of line plots in the top represents spike normalized Reads Per Million Reads (RPM).

could still find weak binding of ZCWPW1 to a few functional sites in the genome including Transcription Start Sites (TSS), Transcription End Site (TES) and CpG islands (CGI). Similarly, Huang and colleagues showed that ZCWPW1 is recruited to some promoter and CGI sites (*Huang et al., 2019*). However, Huang and colleagues found minimal gene dysregulation in *Zcwpw1*-null mice, including little changes in genes whose promoters were bound by ZCWPW1. Wells et al. also found ZCWPW1 recruitment to CGIs when overexpressed in 293T cells, but the binding at these sites was considerably weaker than to PRDM9-dependent hotspots (*Wells et al., 2019*). We conclude from these

studies that ZCWPW1 may bind weakly to some promoters and CGIs which is enhanced by overexpression, but at physiological expression levels, ZCWPW1 is primarily recruited to dually marked (H3K4me3/H3K36me3) sites in a PRDM9-dependent manner. Based on these findings and the rapid evolution of PRDM9 DNA binding zinc finer array, we think it is unlikely that ZCWPW1 plays an important role as transcriptional regulator during meiosis.

Against our expectations, ZCWPW1 is dispensable for initiating DSBs at PRDM9-bound hotspots, as DSBs in *Zcwpw1* KO male mice are still located at PRDM9 bound sites. Our data instead suggests that ZCWPW1 is critical for the efficient repair of these PRDM9-dependent breaks. This conclusion is supported by the partial asynaptic pachytene-like meiotic arrest observed in *Zcwpw1* KO, which coincides with accumulation of γ-H2AX and DMC1, both characterizing failed DSB repair. The findings that PRDM9 histone methyltransferase activity is essential for DSB formation at PRDM9 binding sites (*Diagouraga et al., 2018*; *Powers et al., 2016*) would therefore seem to suggest that a yet to be described PRDM9 histone methyl reader may link PRDM9 to the DSB machinery upstream of ZCWPW1. ZCWPW2, a paralog of ZCWPW1 that diverged early in evolution, is a possible candidate.

In *Zcwpw1* KO spermatocytes, the END-seq profile at hotspots demonstrates the presence of a central peak comparable to WT mice. Such central peaks are lost in *Dmc1* KO mice which fail to undergo strand invasion (*Paiano et al., 2020*). This result might suggest that strand invasion is intact in *Zcwpw1* KO. However, because the asynapsis in arrested spermatocytes in *Zcwpw1* KO is only partial, it is difficult to infer the source of the central peak of END-seq in a heterogenous population of synapsed/asynapsed homologous chromosomes. Therefore, we conclude that ZCWPW1 is a critical factor for DSB repair, and it might act either upstream or downstream of strand invasion step in DSB repair. We propose a model in which ZCWPW1 functions as a PRDM9 specific synapsis factor that is necessary to bind both the cut and uncut homologs to link PRDM9-bound DNA loops to the SC, to facilitate later steps in meiotic recombination. This model is supported by a recent study which suggested that symmetrical PRDM9 binding is critical for meiotic recombination. But how exactly does ZCWPW1 facilitates such synapsis and repair? A hint may lie within the moderately conserved SCP1 domain found at the N-terminus of mouse ZCWPW1 which shares homology to a region of the central axis protein SYCP1 that makes a coiled-coil and facilitates dimerization (*Seo et al., 2016*). Thus, PRDM9 bound loops could be directly tethered to the SC via ZCWPW1. Further studies are needed to explore potential interactions of ZCWPW1 with chromosomal axis and SC components.

In this study we have developed and applied two novel, independent and sensitive methods for mapping meiotic hotspots; END-seq, and ZCWPW1 CUT&RUN. These methods have distinct advantages over previous methods, that include SPO11 oligo sequencing (*Lange et al., 2016*) and SSDS coupled with DMC1 ChIP-seq (*Khil et al., 2012*). END-seq is a method of directly sequencing DSBs, and we have applied it to a single mouse testis to map thousands of hotspots. This method could therefore be applied to virtually any organism for mapping hotspots as it does not require the use of antibodies. ZCWPW1 CUT&RUN also maps mouse hotspots, by mapping the occupancy of the factor ZCWPW1 that is directly downstream of PRDM9-induced histone methylation marks using a polyclonal antibody. Importantly, ZCWPW1 occupancy correlates positively with the strength of hotspots in mice. We applied ZCWPW1 CUT&RUN to 300,000 bulk testicular cells from adult mice (a tiny fraction of the material obtained from a single adult testis) to sensitively map thousands of mouse hotspots. Thus END-seq provides a greater breadth of possible uses including non-model organisms, whereas ZCWPW1 CUT&RUN provides greater flexibility with lower cell numbers in mice. However, both methods currently fall short in detecting the weakest DSB hotspots which could be mapped by SPO11 oligo sequencing and DMC1 SSDS. With optimizations to both techniques, they could be potentially pushed for improved sensitivity for weaker hotspots with low starting cell numbers. This would allow hotspot mapping for currently challenging situations like mouse females or in species with low meiotic cell numbers.

In yeast, plants and at least some vertebrates (those that lost PRDM9 like canids and birds); meiotic hotspots are located in the nucleosome free regions at gene promoters. However, within species that evolved PRDM9, hotspots are specified by the DNA binding activity of the PRDM9 zinc finger array. Therefore, the emergence of PRDM9 was a landmark that re-shaped patterns of meiotic recombination during evolution. The PRDM9-derived pattern of hotspot selection provided more

flexibility in hotspot evolution compared to the ancestral, fixed pattern of hotspots found at promoters and within genes by decoupling hotspot selection away from functional genetic elements.

In this work, we identified ZCWPW1 as an essential factor in the PRDM9 hotspot selection system. This system evolved by co-emergence of (i) a histone writer (PRDM9) which catalyzes formation of H3K4me3 and H3K36me3 dual marks and (ii) a histone methylation reader (ZCWPW1) which recognizes these dual marks to facilitate DSB repair at the PRDM9 bound sites. The presence of these two factors is crucial for hotspot selection and successful meiotic recombination in mice.

# Materials and methods

## Experimental model and subject details

### Mouse models

*Zcwpw1* knockout mouse line (*C57BL/6N-Zcwpw1em1(IMPC)Tcp*) was made as part of the KOMP2-Phase2 project at The Centre for Phenogenomics and was purchased from the Canadian Mouse Mutant Repository. F1 B6/CAST hybrid mouse was generated from mating male CAST (CAST/EiJ) and female B6 (C57BL/6J) (Jackson Laboratory). All experiments were done on adult mice (≥2 weeks of age). Mice were sacrificed for testes dissection. Mice breeding, maintenance and experiments were done according to NIH guidance for the care and use of laboratory animals.

### Protein expression and purification

The mouse ZCWPW1 fragment containing the zinc finger, CW domain, and PWWP domain (residues 1–440; pXC2085) was cloned into a pet28b vector with an N-terminal 6xHis-SUMO tag and transformed into *E. coli* BL21-Codon Plus (DE3)-RIL (Stratagene). An overnight culture was grown in MDAG media, from which 3 L of LB Broth were inoculated and incubated in shaker at 37°C until $A_{600}$ of 0.6 was reached. The temperature was then lowered to 16°C and 30 min later 1 mM $ZnCl_2$ was added with subsequent induction by 0.4 mM isopropyl β-D-1-thiogalactopyranoside (IPTG) 30 min later. Cells were harvested after an overnight growth and immediately suspended in 20 mM Tris (pH 8), 500 mM NaCl, 5% glycerol (v/v), 0.5 mM Tris(2-carboxyethyl)phosphine hydrochloride (TCEP), and 0.1 mM phenylmethanesulfonyl fluoride (PMSF). Cells were lysed by sonification and clarified by centrifugation 47, 850 x g for 1 hr and passed through 3 µm syringe driven filter unit.

The clarified lysate containing mZCWPW1 was then loaded onto a 5 mL His-Trap HP column (GE Healthcare) using NGC chromatography system (BioRad), washed and eluted over a linear gradient from 40 mM to 500 mM Imidazole. Fractions containing ZCWPW1 were pooled and subjected to an overnight digest by ULP1 protease (purified in-house) at 4°C to remove the 6xHis-SUMO N-terminal tag. The cleaved ZCWPW1 was then diluted to 150 mM NaCl using the aforementioned buffer containing no NaCl. This diluted sample was then loaded onto a 5 mL Hi-Trap Q HP column (GE Healthcare) and eluted over a linear gradient of 150 mM to 1000 mM NaCl with the protein eluting at 225 mM NaCl. Fractions with the highest purity were pooled and concentrated via a 10, 000 MWCO centrifugal filter unit (Millipore) to 4.6 mg/mL and flash frozen in liquid $N_2$.

### Antibody production

Rabbit polyclonal antibody for full-length mouse ZCWPW1 was made by GenScript. Rabbits were immunized with the full-length ZCWPW1 recombinant protein, and serum collected after third immunization. Serum was then affinity purified against the full-length ZCWPW1 protein.

### Antibody cross reactivity test

Recombinant ZCWPW1 and ZCWPW2 proteins were produced by cloning coding DNA sequences into vector pET-30a (+) with His tag for protein expression in *E. coli* (GenScript). To test anti-ZCWPW1 cross reactivity with ZCWPW2, recombinant ZCWPW1 and ZCWPW2 were loaded in SDS-PAGE gel and transferred to PVDF membranes. Membranes were then subjected for both Ponceau S staining and immunoblotting with anti-ZCWPW1 rabbit polyclonal antibody.

## Peptide binding experiments

Isothermal Titration Calorimetry (ITC) experiments were conducted at 25°C with a MicroCal PEAQ-ITC automated system (Malvern). H3(1–18)K4me3, H3(21–44)K36me3, and H3(1–44)K4me3K36me3 peptides were ordered from Biomatik. Binding experiments were performed with protein in the sample cell and peptides were injected into the cell with a syringe. H3(1–18) K4me3 experiments were performed with 358 µM peptide in the syringe, and 50 µM protein in the cell. H3(21–44)K36me3 experiments were performed with 600 µM peptide in the syringe and 50 µM protein in the cell. H3 (1–44)K4me3K36me3 experiments were performed with 343 µM peptide in the syringe and 25 µM protein in the cell. The ITC experiments were all conducted with a reference power of 10 µcal/s, with 2.5 µL injections stirred at 750 rpm. The injections were performed over 4 s and with 200 s intervals to allow for equilibrium to be reached in the buffer consisted of 20 mM Tris (pH 8), 75 mM NaCl, 5% glycerol (v/v), and 0.5 mM TCEP. Binding constants were obtained by fitting data to the 'one set of sites' model in the ITC analysis module.

Biotin-Streptavidin Pulldown Assays were conducted using C-terminal biotinylated peptides and streptavidin beads. H3(1–21) and H3(21–44) peptides were ordered from Anaspec, and H3(1–43) peptides were obtained from EpiCypher. Binding reactions consisting of protein (13 µg), biotinylated peptides (0.5 µg), and streptavidin beads were conducted using the binding buffer of 20 mM Tris (pH 8), 75 mM NaCl, 5% glycerol (v/v), 0.5 mM TCEP and 0.1% Triton X-100 overnight at 4°C on a tabletop shaker. Samples were then washed five times with the binding buffer, resuspended in 10 µL SDS loading dye, and heated at 90°C for 5 min before being run on a stain free gel and imaged by GelDoc imager (BIO-RAD).

## CUT&RUN sequencing

To map ZCWPW1 chromatin binding in spermatocytes, Cleavage Under Targets and Release Using Nuclease (CUT&RUN) was performed in accordance to the previously published protocol (*Skene et al., 2018*). To prepare single cell suspension, testes were dissected and chopped after tunica removal, then incubated for 30 min at 37° C in 3 ml dissociation buffer (DMEM + 2 U/ml Dispase (Worthington) + 250 U/ml Collagenase Type 1 (Worthington) + 50 µg/ml DNase I (Sigma)). Dissociation enzymes were inactivated by adding DMEM containing 10% FBS. 300,000 cells were used for each CUT&RUN reaction. Initially, cells were washed twice with CUT&RUN wash buffer (20 mM HEPES (pH 7.5), 150 mM NaCl, 0.5 mM Spermidine, 1X cOmplete EDTA-free Protease Inhibitor Cocktail (Sigma)). Cells were then resuspended in 1 ml wash buffer, and 10 µl of Concanavalin A–coated beads (Bangs Laboratories) suspended in binding buffer (20 mM HEPES-KOH (pH 7.9), 10 mM KCl, 1 mM $CaCl_2$, 1 mM $MnCl_2$) were added to coat the cells with the beads (Concanavalin A–coated beads were activated by pre-washing twice in binding buffer). Cells-beads mixture were incubated with rotation at room temperature (RT) for 10 min in Eppendorf tubes. Thereafter, tubes were placed on magnetic stand until the solution turns clear, and the liquid was discarded. Cells were then resuspended in 50 µl of the antibody buffer (Target-antigen antibody + wash buffer + 0.025% Digitonin + 2 mM EDTA) and incubated at 4°C for 1 hr with rotation. Antibodies' concentrations used were 10 µg/ml, 2 µg/ml and 10 µg/ml for ZCWPW1, H3K4me3 and GFP respectively. After incubation, cells were washed twice via magnet stand using digitonin buffer (wash buffer + 0.025% digitonin) and resuspended in 50 µl digitonin buffer subsequently. 2.5 µl of Protein A–micrococcal nuclease (pA-MN) fusion protein was then added to the cells (final concentration of ~700 ng/ml) and mixed with rotation at 4°C for 1 hr. Unbound pA-MN was removed by washing twice using digitonin buffer. Then cells were resuspended in 150 µl digitonin buffer, and incubated on ice for 5 min to bring samples' temperature to 0°C. Thereafter, 3 µl $CaCl_2$ (100 mM stock solution) was added to activate pA-MN micrococcal nuclease activity, and samples were incubated in ice (0°C) for 30 min. Subsequently, nuclease reaction was stopped by adding 100 µl of 2X stop buffer (340 mM NaCl, 20 mM EDTA, 4 mM EGTA, 0.05% Digitonin, 100 µg/ml RNase A, 50 µg/ml Glycogen). To release CUT&RUN DNA fragments, samples were incubated for 10 min at 37°C. Finally, samples were centrifuged at 16,000 g for 5 min at 4°C and were then placed on magnetic stands to collect supernatant (~250 µl) containing released DNA fragments. To extract DNA, 2.5 µl of 10% SDS and 1.875 µl Proteinase K (20 mg/ml) were added to the released DNA, with incubation and shaking at 65°C for 35 min. To precipitate DNA, 25 µl 3M Sodium Acetate, 2 µl glycoblue (15 mg/ml) (Thermo Fisher) and 687 µl cold 100% ethanol were added, and samples incubated at −20°C overnight, and then

centrifuged in 4°C at maximum speed for 20 min. Supernatant was then removed, and DNA pellet rinsed with 70% ethanol, followed by another centrifugation. Ethanol was removed and the pellet resuspended in 10–50 µl TE buffer. DNA concentration was measured by Qubit (Thermo Fisher). For each sample, library prep was done with 3 ng of DNA using SMARTer ThruPLEX DNA-Seq Kit (Takara) according to manufacturer's protocol. Fourteen cycles of PCR were used in library amplification step, with short elongation time of 10 s to selectively amplify for short fragment expected from CUT&RUN. Paired-end sequencing was done using HiSeq 2500.

## CUT&RUN data analysis

Paired-end reads were mapped to mm10 genome assembly by BWA-MEM version 0.7.17 (*Li and Durbin, 2009*) using default parameters. For subsequent analysis only reads with q-value >30 were used. Peaks were called by MACS2 version 2.1.2 (*Zhang et al., 2008*) using the following parameters: (-p 0.001 –broad-cutoff 0.001 –broad). Read coverage bigwig files were generated by bamCoverage tool of deepTools suite (*Ramírez et al., 2014*) with the options: –MNase –centerReads. Heatmap plotting was done by computeMatrix and plotHeatmap tools. To calculate coverage depth, fragment ends were defined from paired-end reads and only fragments with 130 bp - 200 bp length are considered. Only the central three nucleotides of each fragment are counted and calculated in 50 bp bins. For plotting at functional regions in the genome, we used Transcription Start Sites (TSS), Transcription End Sites (TES) and CpG islands (CGI). The TSS used are all transcript start points from GENCODE release M20. TSSs within ±500 bp are merged and recentered (n = 72,124). The TES used are all transcript end points from GENCODE release M20. TESs within ±500 bp are merged and recentered (n = 87,127). CpG islands (CGIs) were obtained from http://www.haowulab.org/software/makeCGI/model-based-cpg-islands-mm9.txt. We used UCSC liftover to convert to mm10 coordinates (n = 74,692). Blacklisted regions were obtained from https://github.com/Boyle-Lab/Blacklist/blob/master/lists/mm10-blacklist.v2.bed.gz. Integrative Genomics Viewer (IGV) (*Robinson et al., 2011*) was used for read coverage visualization.

For correlation of ZCWPW1 CUT&RUN and SPO11 oligos intensity with other metrics of recombination at DSB hotspots, only autosomal hotspots that coincide with a ZCWPW1 peak were used. CUT&RUN strength for ZCWPW1 is calculated as the sum of in-peak sequencing reads at each hotspot. SPO11-oligo density was used from GSE84689, and strength for DMC1 SSDS and H3K4me3 12dpp are taken from GSE35498. The strength of H3K36me3 (GSE93955), PRDM9 (GSE93955) and H3K4me3 Zygonema (GSE121760) are calculated as the sum of all reads at the SSDS-defined hotspot.

## ZCWPW1 and PRDM9 motif calling

Motifs calling for ZCWPW1 chromatin bound sites was done by HOMER (*Heinz et al., 2010*) version 4.10.4. First, ZCWPW1 CUT&RUN reads were mapped to mm10 genome assembly by bowtie2 (*Langmead and Salzberg, 2012*) version 2.3.5 using the options: –local –very-sensitive-local –no-unal –no-mixed –no-discordant –phred33 -I 10 -X 700. HOMER tag directories were made from mapped bam files by makeTagDirectory command with the option: -tbp 1, and those directories were used to generate homer format peaks by findPeaks command, with the option: -style histone. Motif analysis for ZCWPW1 known motifs and de novo ones was performed by findMotifsGenome.pl command with the options: -size 200 -mask -S 5 -len 14,16,18.

To call motifs of PRDM9, previously published peaks from ChIP-seq of PRDM9[dom2] and PRDM9[Cast] alleles (GEO: GSE93955 *Grey et al., 2017*) were used as input for HOMER findMotifsGenome.pl command with the options: -size 200 -mask -S 5 -len 14,16,18.

## ZCWPW1 expression in mouse tissues

To check for differential ZCWPW1 expression in mouse tissues, *Zcwpw1* WT or *Zcwpw1* KO mice were sacrificed and dissected. All isolated tissues were washed in PBS and subsequently dissociated to single cells as described for CUT&RUN. Cells were then lysed at 4°C on gentle nutation for 1 hr in lysis buffer (50 mM Tris HCl pH7.5, 150 mM NaCl, 1.5 mM MgCl2, 0.2% NP-40, 10% glycerol), supplemented with 1X cOmplete EDTA-free Protease Inhibitor Cocktail (Sigma) and 250 U/µL benzonase (Sigma). After determining protein concentration, 30 µg of protein lysate were used for Western blotting. SDS PAGE was performed using NuPAGE 4–12% Bis-Tris Protein Gels in MES

buffer (ThermoFisher) and proteins were transferred to nitrocellulose membranes. Membranes were subjected to Ponceau staining to evaluate protein content for each sample and transfer efficiency, blocked for 1 hr RT in 2% milk in TBST and blotted using 1:1000 custom anti-ZCWPW1 rabbit polyclonal antibody and 1:1000 anti-GAPDH antibody (NB300-327, Novus Biologicals) in 2% milk in TBST, overnight at 4°C on gentle nutation. Proteins were detected by enhanced chemiluminescence using a c600 Azure imager (Azure Biosystems).

## Hematoxylin and eosin (H&E) staining
Testes were dissected from 8 weeks old adult mice and fixed in 10% formalin. Tissues were embedded in paraffin wax and sectioned at 5 μm thickness, then hematoxylin and eosin (H&E) staining was performed (American Histo Labs).

## Chromosome spreads from spermatocytes
Testes were dissected and decapsulated from 8 weeks old adult mice in 60 mm dish. 200–800 μl of 100 mM sucrose solution were added, followed by gentle chopping and pipetting to dissociate the tissues, and then passed through 70 μm cell strainer to get single cell suspension. Glass slides were pre-cleaned with isopropanol, and then dipped in fixation buffer (1% paraformaldehyde, 0.15% Triton X100, 0.3 mM NaBorate – pH = 8.5). After slide removal from fixation buffer, while small volume of buffer (~5 μl) was kept on the slide,~20 μl of cells in sucrose were added to the slide while tilting it slowly to spread over the cells on the surface. Slides were then incubated in humid tray for 1 hr at room temperature, followed by air drying. Finally, slides were dipped into 0.04% Photoflo (Kodak), dried and saved at –80°C.

## Immunofluorescence staining
Chromosome spread slides were thawed from –80°C, washed in PBT (PBS + 0.1% Tween-20) and blocked in blocking solution (PBT + 0.15% BSA) at room temperature for 1 hr. After three washes in PBT, primary antibody (diluted in blocking solution, 1/50 for anti-DMC1 and 1/200 for anti-SYCP1, anti-SYCP3 and anti-γ-H2AX) incubated overnight at room temperature in humid chamber. Subsequently, slides washed three times and secondary anti-mouse Alexa Fluor 488 and anti-rabbit Alexa Fluor 555 antibodies (1/200 dilution) were incubated at room temperature in humid chamber for 1 hr. Slides were then washed in PBS, counter stained in DAPI and mounted. A Leica DM6000 B microscope was used for imaging. Image analysis was done using Fiji software (*Schindelin et al., 2012*).

## END-seq
END-seq was performed exactly as previously described by *Paiano et al., 2020*.

### Mouse testicular cell isolation
Testes were dissected from 13 weeks adult male mice and placed into a 6 cm tissue culture dish containing DMEM. Tunica albuginea were removed under a microscope, and tubules were gently dissociated with forceps and placed into 50 mL tube containing 20 mL DMEM. After tubules settled to the bottom of the tube, DMEM was aspirated and replaced with 20 mL DMEM containing 0.5 mg/mL Liberase TM (Roche, 5401127001) and incubated at 32°C for 15 min at 500 rpm. Tubules were washed once with fresh DMEM, replaced with 20 mL DMEM containing 0.5 mg/mL Liberase TM and 100 U DNase I (ThermoFisher, EN0521), and incubated at 32°C for 15 min at 500 rpm. Tubules were disrupted by gentle pipetting and passed through a 70 μm Nylon cell strainer (Falcon) repeatedly until tissue debris was fully removed. Cells were pelleted at 1500 rpm at 4°C for 5 min and washed with 10 mL DMEM. Cells were filtered through a 40 μm Nylon cell strainer (Falcon) repeatedly until debris was fully removed and pelleted at 1500 rpm at 4°C for 5 min. Cells were resuspended in 1 mL PBS and counted.

### Embedding cells into agarose plugs
Single-cell suspensions of bulk testicular cells were immediately embedded after isolation into 0.75% agarose plugs. After isolation, cells in 1 mL PBS were diluted to 5–7 million bulk cells/mL PBS and separated into 1 mL of cells per 1.5 mL tube for plug making. Spike-in cells were added at 5% of bulk cell number per tube/plug. Multiple plugs were made per sample if necessary, depending on

number of mice and total cell number isolated, processed in the same tube, and DNA later combined after plug melting. A detailed description for embedding cells into agarose plugs and general END-seq procedure can be found in *Canela et al., 2019* and *Canela et al., 2017*. Briefly, agarose embedded cells were immediately lysed and digested with Proteinase K after agarose solidification. Plugs were then washed with TE, treated with RNase, and stored at 4°C for no longer than one week before the next series of enzymatic reactions.

## Enzymatic reactions
Plugs were treated with sequential combination of Exonuclease VII (NEB) for 1 hr at 37°C followed by Exonuclease T (NEB) for 45 min at 24°C to blunt DNA ends before Illumina adapter ligation (*Canela et al., 2019*). Subsequent steps of A-tailing, adapter ligation, plug melting, chromatin shearing, and second round of adapter ligation for sequencing were performed exactly as previously described (*Canela et al., 2019*; *Canela et al., 2017*).

## END-seq data analysis
### Mapping
Reads were aligned to the mouse (GRCm38p2/mm10) genome using Bowtie version 1.1.2 (*Langmead et al., 2009*) and three mismatches were allowed and the best strata for reads were kept with multiple alignments (-n 3 k 1 l 50). Functions ''view'' and ''sort'' of samtools (*Li, 2011*; *Li et al., 2009*) (version 1.6) were used to convert and sort the mapping output to sorted bam file.

### Peak calling
Peaks were called using MACS 1.4.3 (*Zhang et al., 2008*). END-seq peaks were called using the parameters: `—shiftsize=1000`, `—nolambda`, `—nomodel` and `—keep-dup=all`. Peaks with >2.5 fold-enrichment are kept and those within blacklisted regions (https://sites.google.com/site/anshulkundaje/projects/blacklists) were filtered.

### Spike-in normalization
To normalize END-seq signal from different experiments, we added a spike-in control into END-seq samples which consists of a G1-arrested Ableson-transformed pre-B cell line (*Lig4*$^{-/-}$) carrying a single zinc-finger-induced DSB at the TCRβ enhancer. This site is expected to break in all spike-in cells, which were mixed in at a 5% frequency with bulk testicular cells. END-seq signal was calculated, as RPKM, within ±3 kb window around all hotspot centers. Total intensity was divided by the signal around the spiked-in breaks and then divided by 20 since the spiked-in was added at a 1:20 ratio (5%).

## Single-cell PRDM9 co-expression analysis
To identify *Prdm9* co-expressed genes during meiosis at single cell level, a published dataset for single-cell RNA-sequencing (GEO: GSE107644; *Chen et al., 2018*) was analyzed using R version 3.6.1 with the R packages scran version 1.12.1 and scater version 1.12.2. Unique Molecular Identifiers (UMI) counts were normalized, then pair correlation analysis was done to calculate Spearman's correlation coefficient (Rho) for *Prdm9* co-expression.

## ZCWPW1 evolution analysis
ZCWPW1 orthologs were retrieved from three databases: NCBI HomoloGene, Ensembl orthologs and OrthoDB (*Kriventseva et al., 2019*). Protein sequences of these orthologs were screened for zf-CW and PWWP domains by ScanProsite tool (*de Castro et al., 2006*) and NCBI's conserved domain database (*Marchler-Bauer et al., 2017*). Any ortholog from these databases lacking one of the two domains was excluded. We extended our analysis further to find new ZCWPW1 orthologs by using BLAST alignment approach. We did search in both NCBI protein (nr) and nucleotide (nt) databases via BLAST blastp and tblastn tools respectively, with maximum target sequences of 10,000. For blastp search, we used zf-CW and PWWP domain region from mouse ZCWPW1 (190–234 aa) as query. We then used ScanProsite and CD search to exclude any hit lacking zf-CW or PWWP domain. For tblastn search, we used two queries of zf-CW domain (241–295 aa) and PWWP domain (308–374 aa) in NCBI nt database. We then looked for tblastn hits (accessions) which (a) align to both query

sequences (i.e. zf-CW and PWWP), (b) aligned regions show conserved domain sequence of both zf-CW and PWWP, (c) the distance between the hit's region aligned to zf-CW and the hit's region aligned to PWWP does not exceed 50 kb (in case of alignment to genomic regions). Thereafter, we pooled blastp (nr) and tblastn (nt) results and filtered out any accession for a non-vertebrate species or species with known ZCWPW1 ortholog from databases retrieval. The remaining candidate hits represented either ZCWPW1 or ZCWPW2 orthologs. We used phylogenetic analysis to differentiate between ZCWPW1 and ZCWPW2 orthologs. First, we added known orthologs for both proteins (ZCWPW1 and ZCWPW2) to our candidate sequences from BLAST, and we did multiple sequence alignment using CLC Workbench version 8 (progressive alignment algorithm). We used mouse NSD2 as outgroup sequence in our alignment. Next, we built maximum likelihood phylogenetic tree with 1000 bootstrap replicates. BLAST hits which clustered with ZCWPW1 are considered as novel orthologs in our analysis.

## Multiple sequence alignment for ZCWPW1 orthologs

Amino acid sequences of ZCWPW1 orthologs retrieved from databases (174 species) were used for alignment. For species with multiple isoforms, the longest isoform was used. Alignment was performed by CLC Genomics Workbench using MUSCLE algorithm (*Edgar, 2004*). For graphical representation of the alignment (including conservation and variability scores), all alignment segments corresponding to gaps in the reference sequence (mouse ZCWPW1) were removed. Variability scores were calculated by Protein Variability Server (PVS) (*Garcia-Boronat et al., 2008*) using Shannon entropy analysis (*Shannon, 1948*).

## Data and code availability

All data have been deposited to GEO with the accession number GSE139289. Code used for CUT&RUN data analysis is available at DOI: 10.5281/zenodo.3745123 (*Mahgoub, 2020*).

# Acknowledgements

We would like to thank Kevin Brick for help in analyzing CUT&RUN data and providing analysis code. We also thank Kevin Brick, Florencia Pratto, and Dan Camerini-Otero for help with mice phenotyping, and for their critical comments and discussion on the manuscript. We thank Gang Cheng for providing the F1 hybrid mouse. We thank Rajan Kumar Choudhary and Charmi Mehta for help with cloning and initial purification. We thank Pedro Rocha and Sarah Frail for providing pA-MN and for their help in CUT&RUN optimization. We also thank Steve Coon and Tianwei Li from the NICHD Molecular Genomics Core for NGS support. This work was supported by grants from the National Institutes of Health 1ZIAHD008933 (TSM), GM114306 (XC) and CPRIT RR160029 (XC).

# Additional information

## Funding

| Funder | Grant reference number | Author |
|---|---|---|
| National Institutes of Health | 1ZIAHD008933 | Todd S Macfarlan |
| National Institutes of Health | GM114306 | Xiaodong Cheng |
| National Institutes of Health | CPRIT RR160029 | Xiaodong Cheng |
| National Institutes of Health | Intramural Research Program | André Nussenzweig |

The funders had no role in study design, data collection and interpretation, or the decision to submit the work for publication.

## Author contributions

Mohamed Mahgoub, Conceptualization, Data curation, Software, Formal analysis, Validation, Investigation, Visualization, Methodology, Writing - original draft, Project administration, Writing - review and editing; Jacob Paiano, Data curation, Formal analysis, Validation, Investigation, Visualization,

Methodology, Writing - review and editing; Melania Bruno, Validation, Investigation, Visualization, Methodology, Writing - review and editing; Wei Wu, Data curation, Software, Formal analysis, Visualization, Writing - review and editing; Sarath Pathuri, Xing Zhang, Investigation, Visualization; Sherry Ralls, Resources, Investigation, Project administration; Xiaodong Cheng, André Nussenzweig, Supervision, Funding acquisition, Methodology, Writing - review and editing; Todd S Macfarlan, Conceptualization, Resources, Supervision, Funding acquisition, Validation, Investigation, Visualization, Methodology, Writing - original draft, Project administration, Writing - review and editing

### Author ORCIDs
Mohamed Mahgoub (iD) https://orcid.org/0000-0003-2687-7874
Melania Bruno (iD) http://orcid.org/0000-0002-8401-7744
André Nussenzweig (iD) http://orcid.org/0000-0002-8952-7268
Todd S Macfarlan (iD) https://orcid.org/0000-0003-2495-9809

### Ethics
Animal experimentation: All mice experiments were done in accordance with NIH approved animal study protocol (ASP18-026).

### Decision letter and Author response
Decision letter https://doi.org/10.7554/eLife.53360.sa1
Author response https://doi.org/10.7554/eLife.53360.sa2

## Additional files

### Supplementary files
• Transparent reporting form

### Data availability
All data have been deposited to GEO with the accession number GSE139289.

The following dataset was generated:

| Author(s) | Year | Dataset title | Dataset URL | Database and Identifier |
|---|---|---|---|---|
| Mahgoub M, Paiano J, Bruno M, Wu W, Pathuri S, Zhang X, Ralls S, Cheng X, Nussenzweig A, Macfarlan T | 2020 | Dual Histone Methyl Reader ZCWPW1 Facilitates Repair of Meiotic Double Strand Breaks | https://www.ncbi.nlm.nih.gov/geo/query/acc.cgi?acc=GSE139289 | NCBI Gene Expression Omnibus, GSE139289 |

The following previously published datasets were used:

| Author(s) | Year | Dataset title | Dataset URL | Database and Identifier |
|---|---|---|---|---|
| Brick K, Thibault-Sennett S, Smagulova F, Lam KW, Pu Y, Pratto F, Camerini-Otero RD, Petukhova GV | 2018 | Extensive sex differences at the initiation of meiotic recombination | https://www.ncbi.nlm.nih.gov/geo/query/acc.cgi?acc=GSE99921 | NCBI Gene Expression Omnibus, GSE99921 |
| Brick K, Smagulova F, Khil P, Camerini-Otero R, Petukhova G | 2012 | Genetic recombination is directed away from functional genetic sites in mice | https://www.ncbi.nlm.nih.gov/geo/query/acc.cgi?acc=GSE35498 | NCBI Gene Expression Omnibus, GSE35498 |
| Lam KG, Brick K, Cheng G, Pratto F, Camerini-Otero RD | 2019 | Cell-type-specific genomics reveals histone modification dynamics in mammalian meiosis | https://www.ncbi.nlm.nih.gov/geo/query/acc.cgi?acc=GSE121760 | NCBI Gene Expression Omnibus, GSE121760 |
| Grey C, Clément | 2017 | In vivo binding of PRDM9 reveals | https://www.ncbi.nlm. | NCBI Gene |

| | | | | | |
|---|---|---|---|---|---|
| JA, Buard J, Le-blanc B, Gut I, Gut M, Duret L, de Massy B | | interaction with non-canonical genomic sites | | nih.gov/geo/query/acc. cgi?acc=GSE93955 | Expression Omnibus, GSE93955 |
| Altemose N, Myers SR, Hatton E, Donnelly P | 2016 | Humanized PRDM9 Mouse Testis H3K4me3 and DMC1 ChIP-seq | | https://www.ncbi.nlm. nih.gov/geo/query/acc. cgi?acc=GSE73833 | NCBI Gene Expression Omnibus, GSE73833 |
| Lange J, Keeney S | 2016 | The landscape of mouse meiotic double-strand break formation, processing and repair | | https://www.ncbi.nlm. nih.gov/geo/query/acc. cgi?acc=GSE84689 | NCBI Gene Expression Omnibus, GSE84689 |
| Chen Y, Zheng Y, Gao Y, Lin Z, Yang S, Wang T, Wang Q, Xie N, Hua R, Liu M, Sha J, Griswold M, Li J, Tang F, Tong M | 2018 | Single-cell RNA-seq Uncovers Dynamic Processes and Critical Regulators in Mouse Spermatogenesis | | https://www.ncbi.nlm. nih.gov/geo/query/acc. cgi?acc=GSE107644 | NCBI Gene Expression Omnibus, GSE107644 |
| Brick K, Smagulova F, Pu Y, Camerini-Otero R, Petukhova G | 2016 | The evolutionary dynamics of meiotic recombination initiation in mice | | https://www.ncbi.nlm. nih.gov/geo/query/acc. cgi?acc=GSE75419 | NCBI Gene Expression Omnibus, GSE75419 |

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
