## [Decision Letter]

**Acceptance summary:**

This paper reports the identification of the dual reader of Prdm9 deposited histone modifications in mouse meiosis. This is an important step and missing link in the process of meiotic recombination. This study indicates that this reader is important for efficient DSB repair.

**Decision letter after peer review:**

Thank you for submitting your article "Dual Histone Methyl Reader ZCWPW1 Facilitates Repair of Meiotic Double Strand Breaks" for consideration by *eLife*. Your article has been reviewed by three peer reviewers, including Bernard de Massy as the Reviewing Editor and Reviewer #1, and the evaluation has been overseen by Jessica Tyler as the Senior Editor.

The reviewers have discussed the reviews with one another and the Reviewing Editor has drafted this decision to help you prepare a revised submission.

Summary:

The authors characterise a new protein, ZCWPW1, that is co-expressed with PRDM9 in mouse, a protein that has been shown to specify meiotic SPO11-DSB hotspot locations via the zinc-finger domain, and histone-trimethylation activity. The authors demonstrate in vitro binding of ZCWPW1 to methylated histones. Evidence for co-evolution of ZCWPW1 and PRDM9 is presented. Patterns of ZCWPW1 chromatin binding are assessed (corresponding to PRDM9 binding sites). Phenotypes of ZCWPW1 KO are presented both in terms of immunofluorescence staining of certain factors, and in the assessment of DSB patterns using END-seq, which is used here for the first time to detect DSBs in meiocytes.

This study provides novel and interesting information about meiotic DSB control in mice. Most of the conclusions are convincing, however several interpretations required additional experimental support and the presentation of the data requires revision and clarification.

Importantly, as the authors have submitted their manuscript at the same time as the ones from Wells et al. and Huang et al. , the results from these two papers have to be integrated in the Discussion to provide a more coherent view or to point out discrepancies if any (ie: why no enrichment of ZCWPW1 at TSS is detected in this study whereas it is detected by Huang et al., is any enrichment detected in CpG rich region as suggested by Wells et al?)

Essential revisions:

1) To show the specificity of the antibody for ZCWPW1 vs ZCWPW2.

2) Test affinity with peptides of identical lengths (i.e. H3 (1-44)).

3) Analysis of ZCWPW1 binding: explain data, normalization, analyze throughout the genome including TSS, enhancers, CpG islands, along genes; analyze quantitatively and correlate with other landmarks (H3K4me3, H3K36me3, Spo11, DMC1, PRDM9…), analyse in the PAR (Prdm9 independent DSB sites). Discuss potential impact on transcription activity.

4) Revise cytology of ZCWPW1 KO: provide better quality images, quantify synapsis (i.e. % of synapsis and percentage of cells reaching partial or full synapsis) and a DSB repair marker (gH2AX or RPA or DMC1), and refer to other manuscripts for complementary data.

5) Conclusion of a post strand invasion defect should be toned down, not convincing.

6) Many clarifications issues in the data presented and many inaccuracies and overinterpretations in the text.

---

## [Author Response]

Summary:The authors characterise a new protein, ZCWPW1, that is co-expressed with PRDM9 in mouse, a protein that has been shown to specify meiotic SPO11-DSB hotspot locations via the zinc-finger domain, and histone-trimethylation activity. […] Most of the conclusions are convincing, however several interpretations required additional experimental support and the presentation of the data requires revision and clarification.

We would like to thank the reviewers for their accurate assessment of the conclusions of our manuscript. We provide a detailed response to the reviewers’ thoughtful queries below, which have allowed us to improve our manuscript and better integrate with the studies of Well et al. and Huang et al.

Importantly, as the authors have submitted their manuscript at the same time as the ones from Wells et al. and Huang et al. , the results from these two papers have to be integrated in the Discussion to provide a more coherent view or to point out discrepancies if any (i.e.: why no enrichment of ZCWPW1 at TSS is detected in this study whereas it is detected by Huang et al., is any enrichment detected in CpG rich region as suggested by Wells et al?)

We have added a paragraph discussing ZCWPW1 binding to TSSs and CpG islands, citing results from Huang et al. and Wells et al. (Discussion paragraph three).

We don’t find a discrepancy between Huang et al. results and ours. The percentage of TSS/Promoters bound by ZCWPW1 could vary based on how TSS/Promoter are defined (i.e. database, region length) and parameters of peak calling for ZCWPW1 (see our responses for comments 3 and 36). ZCWPW1 peaks overlapping TSS/promoters are weaker compared to peaks overlapping hotspots (response for comment 3). Furthermore, Huang et al. performed RNA-seq in *Zcwpw1* KO spermatocytes, and found most of the differentially expressed genes were not the genes whose promoters overlapped with ZCWPW1 binding sites. Therefore, we think it is unlikely that ZCWPW1 plays a critical role as transcriptional regulator during meiosis, though that can't be completely ruled out.

Regarding the Wells et al. results, we also observe ubiquitous ZCWPW1 binding to CpG islands (CGI) when over-expressing ZCWPW1 in 293T cells using CUT&RUN (data not shown). But in vivo we find only 5% of ZCWPW1 peaks overlapping with CGIs in spermatocytes (Figure 3—figure supplement 3B), yet there was no significant ZCWPW1 enrichment genome wide at CGIs. In addition, Wells et al. suggest that ZCWPW1 binds chromatin independently of PRDM9, based on over-expression in 293T cells. We believe this is not the case for endogenously expressed ZCWPW1 in spermatocytes. Huang et al. found that endogenously expressed ZCWPW1 mutant (which lacks zf-CW H3K4me3 reader activity) lacks DNA binding affinity. Based on these results in sum, we conclude that ZCWPW1 chromatin binding in spermatocytes is entirely mediated via its histone H3 reading domains, and mainly in a PRDM9-dependent manner.

Essential revisions:1) To show the specificity of the antibody for ZCWPW1 vs ZCWPW2.

We generated purified recombinant ZCWPW1 and ZCWPW2 in bacteria, and performed western blotting using ZCWPW1 antibody. We found no cross reactivity of our ZCWPW1 antibody with ZCWPW2 protein. These results are now shown in the revised manuscript in Figure 1—figure supplement 2.

2) Test affinity with peptides of identical lengths (i.e. H3 (1-44)).

We appreciate the suggestion. In the biotin-streptavidin pulldown assays (Figure 1D), we already used peptides of identical lengths for no-modification, single-modification at either H3K4 or H3K36 and dual-modification at both K4 and K36. We concluded the binding affinities in the order of double > single > no modification. These peptides had to be specially synthesized because they were not commercially available. In the quantitative ITC experiments (Figure 1E), which require much larger amounts of highly purified protein and peptides (that again had to be specially synthesized and purified at large costs), we compared binding of double modification to that of no modification in the context of peptides of identical length H3(1-44). We found no binding to peptide without modification. We thus used shorter peptides (mainly for cost saving, as these were commercially available at the large quantities required for ITC) including methylated K4 or K36 for measuring the binding affinities of single modification. It is exceedingly unlikely that the unmodified portion of peptide would affect the binding.

3) Analysis of ZCWPW1 binding: explain data, normalization, analyze throughout the genome including TSS, enhancers, CpG islands, along genes; analyze quantitatively and correlate with other landmarks (H3K4me3, H3K36me3, Spo11, DMC1, PRDM9…), analyse in the PAR (Prdm9 independent DSB sites). Discuss potential impact on transcription activity.

We made the code and pipeline used for data analysis of ZCWPW1 CUT&RUN publicly available at DOI: 10.5281/zenodo.3740571.

Details of peak calling are given in methods (below). Using these thresholds, ~ 90% of autosomal ZCWPW1 peaks in both B6 and B6xCAST, are at DSB hotspots. Furthermore, both in B6 and B6xCAST, the ZCWPW1 CUT&RUN signal is very weak at the remaining 10%, perhaps indicating that a substantial fraction of these are false-positive peak calls (Author response image 1). ~5 to 9% of ZCWPW1 peaks coincide with “functional” sites (TSS, TES or CGIs); it is possible that ZCWPW1 has weak affinity for these sites, however there is no enrichment relative to the GFP control (Author response image 2 and manuscript newly added Figure 3—figure supplement 3C). Thus, we think that it is unnecessarily speculative to discuss the potential impact on transcription given these results alone. We now include heatmaps of the raw ZCWPW1 signal compared to H3K4me3 and the GFP control at TSS, TES and CGIs for B6 (Figure 3—figure supplement 3A) and for B6/CAST B6 (Author response image 3)

**Author response image 1. sa2fig1:** ZCWPW1 peaks overlapping hotspots are stronger than non-hotspots. Left panel: B6, Right panel B6xCAST.

**Author response image 2. sa2fig2:** ZCWPW1 CUT&RUN shows very little enrichment at functional sites in the genome. Sequencing reads (q > 30) at each set of genomic intervals were counted for our four CUT&RUN experiments as well as for H3K4me3 ChIP-Seq data in 12dpp testis. The genomic intervals used are TSS and TES from GENCODE vM20 (gencodeTSS, gencodeTES), peaks from our CUT&RUN experiments (ZCWPW1 (B6xCST), ZCWPW1 (B6), H3K4me3 (B6), H3K4me3 (B6xCST)), CpG islands (CGI), DSB hotspots (HS (B6), HS (B6XCST)). (**a**) The percentage of total reads in each set of genomic intervals is shown. (**b**) The enrichment relative to GFP CUT&RUN in B6 mice is shown. Color scale in b is scaled by the log of enrichment values.

**Author response image 3. sa2fig3:** ZCWPW1 CUT&RUN shows very little enrichment at functional sites in the genome. Only sequencing reads with a mapping quality of >30 were used. The top and second-to-top panels represent DSB hotspots with / without a called ZCWPW1 peak, respectively. Data here are from B6 x CAST mice.

We have also included scatterplots of ZCWPW1 signal compared to other metrics of meiotic recombination (Author response images 4-5 and manuscript Figure 3—figure supplement 2). ZCWPW1 strength exhibits a positive correlation with H3K4me3, PRDM9 ChIP-seq intensity (Grey et al., 2017), Spo11-oligo density and DMC1-SSDS signal. While ZCWPW1 CUT&RUN signal correlates best with H3K4me3 derived from ChIP-seq in 12dpp mice (Baker et al., 2014), noise in all of these strength estimates precludes us from inferring which of these readouts is truly the closest correlate of ZCWPW1 activity.

**Author response image 4. sa2fig4:** ZCWPW1 CUT&RUN signal correlates with other metrics of recombination at DSB hotspots. Data are shown for ZCWPW1 CUT&RUN in B6 mice. For all panels, only autosomal hotspots that coincide with a ZCWPW1 peak were used. Strength for SSDS and H3K4me3 12dpp are taken from Brick et al., Nature 2018. CUT&RUN strength for H3K4me3 and ZCWPW1 is calculated as the sum of in-peak sequencing reads at each hotspot. The strength of H3K36me3 is calculated as the sum of all reads at the SSDS-defined hotspot. ZCWPW1 signal correlates best with H3K4m3 ChIP-Seq data from 12dpp mice. Notably however, it correlates less well with H3K4me3 from isolated Zygotene spermatocytes. This variation is likely a reflection of variation in ChIP / experimental enrichment and highlights the difficulty of attributing any of these measures as being “best” correlated with ZCWPW1 binding. ZCWPW1 appears to correlate equally well with Spo11-oligo and DMC1-SSDS measures of DSB frequency. Interestingly however, the correlation with DMC1-SSDS is notably improved in mice that lack the ability to repair DSBs (Hop2ko). The correlation with H3K36me3 is relatively lower, however the H3K36me3 signal at hotspots is notably weaker than that of the other metrics. Thus, the low correlation may be strongly influenced by background noise.

**Author response image 5. sa2fig5:** This figure is simply to provide a comparison point to Author response image 3. Spo11-oligo signal correlates with other metrics of recombination. For all panels, only autosomal hotspots that coincide with a ZCWPW1 peak were used. Spo11 signal correlates best with DMC1-SSDS. Notably however, it correlates less well with DMC1-SSDS from female mice. As with ZCWPW1, the correlation with H3K36me3 is poor, since the H3K36me3 signal at hotspots is notably weaker than that of the other metrics.

We also added new Figure 3B showing overlap of ZCWPW1 peaks with other metrics (DMC1, SPO11, H3K4me3, H3K36me3)

We also added new figure showing ZCWPW1 read coverage at the PAR region. We do not observe any enrichment of ZCWPW1 (relative to GFP control) in the PAR or at PRDM9 independent hotspots in the PAR-adjacent region (Figure 3—figure supplement 4). The closest ZCWPW1 peak to the PAR in B6 mice is ~3.5 Mb away.

Peak calling method:

We used MACS2 to call peaks with the following arguments.

-t {ZCWPW1 CUT&RUN BAM file; only reads with q-value >30}

-c {CFP CUT&RUN BAM file; only reads with q-value >30}

-p 0.001

-g mm

-broad-cutoff 0.001

-broad

Alignment of published data:

H3K4me3 ChIP-seq from 12dpp mice (Baker et al., 2014) and PRDM9 ChIP-seq data (Grey et al., 2017) were aligned to the mm10 reference genome using bwa 0.7.12.

4) Revise cytology of ZCWPW1 KO: provide better quality images, quantify synapsis (i.e. % of synapsis and percentage of cells reaching partial or full synapsis) and a DSB repair marker (gH2AX or RPA or DMC1), and refer to other manuscripts for complementary data.

We now provide higher quality images for cytology of spermatocyte spreads. Additionally, we performed quantitative analysis for cytology and added three new figures for: (i) Distribution of meiotic prophase I stages (Figure 5B) (ii) Counts for number of fully synapsed homologous chromosomes (autosomes) per spermatocyte (Figure 5D) (iii) DMC1 foci count per cell (Figure 5F). We added citations for cytology results from other ZCWPW1 papers (M. Li et al., 2019; Wells et al., 2019).

5) Conclusion of a post strand invasion defect should be toned down, not convincing.

We added a new sentence stating that with our current data we can’t exclude ZCWPW1 involvement in homolog strand invasion. It states that "Therefore, we conclude that ZCWPW1 is a critical factor for DSB repair, and it might act either upstream or downstream of strand invasion step in DSB repair"

6) Many clarifications issues in the data presented and many inaccuracies and overinterpretations in the text.

This is a very general statement that cannot be directly addressed. We have clarified all the direct queries raised by the reviewers.